# EFFICIENT PARAMETRIC APPROXIMATIONS OF NEURAL NETWORK FUNCTION SPACE DISTANCE

## ABSTRACT

It is often useful to compactly summarize important properties of a training dataset so that they can be used later without storing and/or iterating over the entire dataset. We consider a specific case of this: approximating the function space distance (FSD) over the training set, i.e. the average distance between the outputs of two neural networks. We propose an efficient approximation to FSD for ReLU neural networks based on approximating the architecture as a linear network with stochastic gating. Despite requiring only one parameter per unit of the network, our approach outcompetes other parametric approximations with larger memory requirements. Applied to continual learning, our parametric approximation is competitive with state-of-the-art nonparametric approximations which require storing many training examples. Furthermore, we show its efficacy in influence function estimation, allowing influence functions to be accurately estimated without iterating over the full dataset.

## 1 INTRODUCTION

There are many situations in which we would like to compactly summarize a model's training data. One motivation is to reduce storage costs: in continual learning, an agent continues interacting with its environment over a long time period — longer than it is able to store explicitly — but we would still like it to avoid overwriting its previously learned knowledge as it learns new tasks (Goodfellow et al., 2013). Even in cases where it is possible to store the entire training set, one might desire a compact representation in order to avoid expensive iterative procedures over the full data. Examples include influence function estimation (Koh & Liang, 2017; Bae et al., 2022a), model editing (De Cao et al., 2021; Mitchell et al., 2021), and unlearning (Bourtoule et al., 2021). While there are many different aspects of the training data that one might like to summarize, we are often particularly interested in preventing the model from changing its predictions too much on the distribution of previously seen data.

Methods to prevent such catastrophic forgetting, especially in the field of continual learning, can be categorized at a high level into parametric and nonparametric approaches. Parametric approaches store the parameters of a previously trained network, together with additional information about how important different directions in parameter space are for preserving past knowledge; the canonical example is Elastic Weight Consolidation (Kirkpatrick et al., 2017, EWC), which uses a diagonal approximation to the Fisher information matrix. Nonparametric approaches explicitly store a collection (coreset) of training examples, often optimized directly to be the most important or memorable ones (Rudner et al., 2022; Pan et al., 2020; Titsias et al., 2019). Currently, the most effective approaches to prevent catastrophic forgetting are nonparametric, since it is difficult to find sufficiently accurate parametric models. However, this advantage is at the expense of high storage requirements.

We focus on the problem of approximating *function space distance (FSD)*: the amount by which the outputs of two networks differ, in expectation over the training distribution. Benjamin et al. (2018) observed that regularizing FSD over the previous task data is an effective way to prevent catastrophic forgetting. Other tasks such as influence estimation (Bae et al., 2022a), model editing (Mitchell et al., 2021), and second-order optimization (Amari, 1998; Bae et al., 2022b) have also been formulated in terms of FSD regularization or equivalent locality constraints. In this paper, we formulate the problem of approximating neural network FSD and propose novel parametric approximations. Our methods significantly outperform previous parametric approximations despite

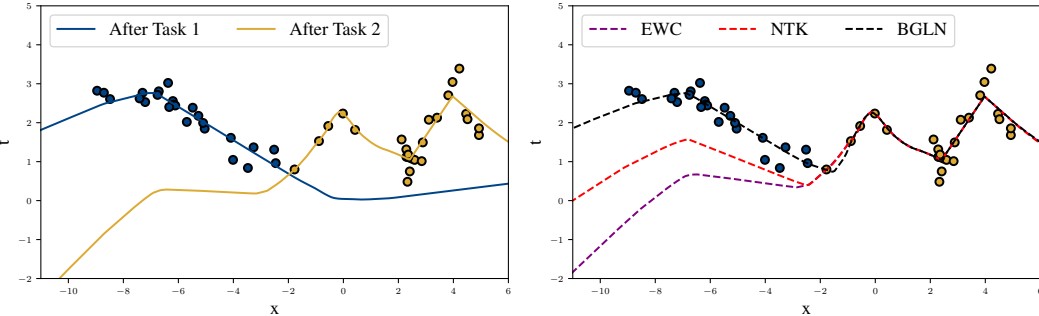

**Figure 1:** Comparison of FSD regularization on a one-dimensional regression task. **(Left)** Training sequentially on two tasks (blue, then yellow), results in catastrophic forgetting. **(Right)** BGLN retains performance on Task 1 after training on Task 2 more accurately than other methods. Note that the $y$-axis represents function space distances for each datapoint.

being much more memory-efficient, and are also competitive with nonparametric approaches to continual learning.

Several parametric approximations, like EWC, are based on a second-order Taylor approximation to the FSD, leading to a quadratic form involving the Fisher information matrix $F_{\theta}$ or some other metric matrix $G_{\theta}$, where $\theta$ denotes the network parameters. Second-order approximations help because one can approximate $F_{\theta}$ or $G_{\theta}$ by sampling vectors from a distribution with these matrices as the covariance. Then, tractable probabilistic models can be fit to these samples to approximate the corresponding distribution. Unfortunately, these tend to be inaccurate for continual learning, especially in comparison with nonparametric approaches. We believe the culprit is the second-order Taylor approximation: we show in several examples that even the exact second-order Taylor approximation can be a poor match to FSD over the scales relevant to continual learning, like average performance over sequentially learned tasks. This is consistent with a recent line of results that find linearized approximations of neural networks to be an inaccurate model of their behavior (Seleznova & Kutyniok, 2022a;b; Hanin & Nica, 2019; Bai et al., 2020; Huang & Yau, 2020).

Our contribution, the *Bernoulli Gated Linear Network (BGLN)*, makes a parametric approximation to neural network FSD which does *not* make a second-order Taylor approximation in parameter space, and hence is able to capture nonlinear interactions between parameters of the network. Specifically, it linearizes each layer of the network with respect to its inputs. In the case of ReLU networks, our approximation yields a linear network with stochastic gating. Linearizing the ReLU function requires computing its gradient, which can be approximated as an independent Bernoulli random variable for each unit. We derive a stochastic as well as a deterministic estimate of the FSD in this setting, both of which rely only on the first two moments of the data.

To demonstrate the practical usefulness of our approximation, we evaluate its closeness to the true empirical FSD. We show that our method estimates and optimizes the true FSD better than other estimators in settings that are prone to forgetting. Further, we show its application and performance in two applications. When applied to continual learning, it outcompetes state-of-the-art methods on sequential MNIST and CIFAR100 tasks, with at least 90% lower memory requirements than nonparametric methods. When applied to influence function estimation, our method achieves over 95% correlation with the ground truth, without iterating over or storing the full dataset.

## 2 BACKGROUND

Let $z = f(\mathbf{x}, \boldsymbol{\theta})$ denote the function computed by a neural network, which takes in inputs $\mathbf{x}$ and parameters $\boldsymbol{\theta}$. Consistent with prior works, we use FSD to refer to the expected output space distance[1] $\rho$ between the outputs of two neural networks (Benjamin et al., 2018; Grosse, 2021; Bae et al., 2022b), with respect to the training distribution, as defined in equation 1. When the population

---

[1]Note that we use the term *distance* throughout since we focus on Euclidean distance in practice. However, other metrics like KL divergence, which are not distances, are also possible and commonly used.

distribution is inaccessible, the empirical distribution is often used as a proxy:

$$D(\boldsymbol{\theta}_0, \boldsymbol{\theta}_1, p_{\text{data}}) = \mathbb{E}_{\mathbf{x} \sim p_{\text{data}}}[\rho(f(\mathbf{x}, \boldsymbol{\theta}_0), f(\mathbf{x}, \boldsymbol{\theta}_1))] \approx \frac{1}{N} \sum_{i=1}^{N} \rho(f(\mathbf{x}^{(i)}, \boldsymbol{\theta}_0), f(\mathbf{x}^{(i)}, \boldsymbol{\theta}_1)), \quad (1)$$

where $\mathbf{x}$ is a random data point sampled from the data-generating distribution $p_{\text{data}}$. Constraining the FSD term has been successful in preventing catastrophic forgetting (Benjamin et al., 2018), computing influence functions (Bae et al., 2022a), training teacher-student models (Hinton et al., 2015), and fine-tuning pre-trained networks (Jiang et al., 2019; Mitchell et al., 2021).

Consider the continual learning setting as a motivating example. Common benchmarks (Normandin et al., 2021) involve sequentially learning tasks $t \in \{1, \ldots, T\}$, using cost function $\mathcal{J}$ and a penalty on the FSD between the learned parameters, $\boldsymbol{\theta}$ and optimal parameters of previous tasks, $\boldsymbol{\theta}_i$, computed over the previously seen data distribution, $p_i$, and scaled by hyperparameter, $\lambda_{\text{FSD}}$.

$$\boldsymbol{\theta}_t = \arg\min_{\boldsymbol{\theta}} \mathcal{J}(\boldsymbol{\theta}) + \lambda_{\text{FSD}} \sum_{i=1}^{t-1} D(\boldsymbol{\theta}, \boldsymbol{\theta}_i, p_i) \qquad (2)$$

Continuing with the notation in equation 1, one way to regularize the (empirical) FSD is to store the training set and explicitly evaluate the network outputs with both $\boldsymbol{\theta}_0$ and $\boldsymbol{\theta}_1$ (perhaps on random mini-batches). However, this has the drawbacks that one would need to store and access the entire training set throughout training (precisely the thing continual learning research tries to avoid) and that the FSD needs to be estimated stochastically. Instead, we would like to compactly summarize information about the training set or the training distribution.

Many (but not all) practical FSD approximations are based on a second-order Taylor approximation:

$$D(\boldsymbol{\theta}_0, \boldsymbol{\theta}_1, p_{\text{data}}) \approx \frac{1}{2}(\boldsymbol{\theta}_1 - \boldsymbol{\theta}_0)^{\top} \boldsymbol{G}_{\boldsymbol{\theta}}(\boldsymbol{\theta}_1 - \boldsymbol{\theta}_0), \qquad (3)$$

where $\boldsymbol{G}_{\boldsymbol{\theta}} = \nabla^2_{\boldsymbol{\theta}} D(\boldsymbol{\theta}_0, \boldsymbol{\theta}, p_{\text{data}})$. In the case where the network outputs parametrize a probability distribution and $\rho$ corresponds to KL divergence, $\boldsymbol{G}_{\boldsymbol{\theta}}$ reduces to the more familiar Fisher information matrix $F_{\boldsymbol{\theta}} = \mathbb{E}_{\mathbf{x} \sim p_{\text{data}}, \mathbf{y} \sim P_{\mathbf{y}|\mathbf{x}}(\boldsymbol{\theta})}[\nabla_{\boldsymbol{\theta}} \log p(\mathbf{y}|\boldsymbol{\theta}, \mathbf{x}) \nabla_{\boldsymbol{\theta}} \log p(\mathbf{y}|\boldsymbol{\theta}, \mathbf{x})^{\top}]$, where $P_{\mathbf{y}|\mathbf{x}}(\boldsymbol{\theta})$ represents the model's predictive distribution over $\mathbf{y}$. It is possible to sample random vectors in parameter space whose covariance is $\boldsymbol{G}_{\boldsymbol{\theta}}$ (Martens et al., 2012; Grosse & Martens, 2016; Grosse, 2021), and some parametric FSD approximations work by fitting simple statistical models to the resulting distribution. For instance, assuming all coordinates are independent gives a diagonal approximation (Kirkpatrick et al., 2017), and more fine-grained independence assumptions between network layers yield a Kronecker-factored approximation (Martens & Grosse, 2015; Ritter et al., 2018). In practice, instead of sampling vectors whose covariance is $\boldsymbol{G}_{\boldsymbol{\theta}}$, many works instead use the empirical gradients during training, whose covariance is the empirical Fisher matrix. We caution the reader that the empirical Fisher matrix is less well motivated theoretically and can result in different behavior (Kunstner et al., 2019).

## 3  A DATA-BASED PARAMETRIC APPROXIMATION

Motivated by the above, we propose BGLN (Bernoulli Gated Linear Network), which approximates a given model architecture as a linear network with stochastic gating and captures more nonlinearities between the parameters than the previous parametric approximations discussed. It is applicable to different architectures, but we first explicitly derive our approximation for multilayer perceptrons (MLPs), with $L$ fully-connected layers and the ReLU activation function, $\phi$. We then discuss and empirically evaluate its generalization to convolutional networks. In Section 3.4, we extend our method to a class-conditioned version which takes into account different classification categories.

For MLPs with inputs $\mathbf{x}$, layer $l$ weights and biases $(\boldsymbol{W}^{(l)}, \boldsymbol{b}^{(l)})$ and outputs $\boldsymbol{z}$, the computation of preactivations and activations at each layer is recursively defined as follows.

$$\boldsymbol{s}^{(l)} = \boldsymbol{W}^{(l)} \boldsymbol{a}^{(l-1)} + \boldsymbol{b}^{(l)}, \ \boldsymbol{a}^{(l)} = \phi(\boldsymbol{s}^{(l)}) \qquad (4)$$

with $\boldsymbol{a}^{(0)} = \boldsymbol{x}$, and $\boldsymbol{s}^{(L)} = \boldsymbol{z}$. We denote $\boldsymbol{z}_0$ and $\boldsymbol{z}_1$ to be the outputs obtained with parameters $\boldsymbol{\theta}_0$ and $\boldsymbol{\theta}_1$, respectively. With Euclidean distance as the output space distance, $\rho$, we can rewrite the FSD as a sum of the first two moments of the difference between the output vectors $\Delta \boldsymbol{z} := \boldsymbol{z}_1 - \boldsymbol{z}_0$.

$$\mathbb{E}_{\mathbf{x} \sim p_{\text{data}}}[\rho(\boldsymbol{\theta}_0, \boldsymbol{\theta}_1; \mathbf{x})] = \mathbb{E}[\frac{1}{2}||\Delta \boldsymbol{z}||^2] = \frac{1}{2}||\mathbb{E}[\Delta \mathbf{z}]||^2 + \frac{1}{2} tr \, \text{Cov}(\Delta \mathbf{z}) \qquad (5)$$

---

**Algorithm 1** BGLN-S

---

**Require:** $\mathbb{E}[\mathbf{x}], \mathrm{Cov}(\mathbf{x}), \{\boldsymbol{W}, \boldsymbol{b}\}_1^{L-1}, \{\mu\}_1^{L-1}$
 1: $\mathbb{E}[\boldsymbol{a}_0], \mathbb{E}[\boldsymbol{a}_1] \leftarrow \mathbb{E}[\mathbf{x}]$
 2: $\mathrm{Cov}(\boldsymbol{a}_0), \mathrm{Cov}(\boldsymbol{a}_1) \leftarrow \mathrm{Cov}(\mathbf{x})$
 3: $\boldsymbol{a}_0, \boldsymbol{a}_1 \sim \mathcal{N}(\mathbb{E}[\boldsymbol{a}_0], \mathrm{Cov}(\boldsymbol{a}_0))$          ▷ sample inputs using first two moments
 4: $\Delta a \leftarrow 0$
 5: **for** $l \leftarrow 1$ to $L-1$ **do**
 6:      $\mathbf{m} \sim Ber(\mu^{(l)})$          ▷ sample Bernoulli random variables
 7:      $\boldsymbol{s}_0 \leftarrow \boldsymbol{W}_0^{(l)}\boldsymbol{a}_0 + \boldsymbol{b}_0^{(l)}$          ▷ linear layers preserve input-output relationship
 8:      $\Delta \boldsymbol{s} \leftarrow \Delta\boldsymbol{W}^{(l)}\boldsymbol{a}_0 + \boldsymbol{W}_1^{(l)}\Delta\boldsymbol{a} + \Delta\boldsymbol{b}^{(l)}$
 9:      $\boldsymbol{a}_0 \leftarrow \mathbf{m} \odot \boldsymbol{s}_0$          ▷ stochastic gating
10:      $\Delta\boldsymbol{a} \leftarrow \mathbf{m} \odot \Delta\boldsymbol{s}$
11:      $\boldsymbol{s}_1 \leftarrow \boldsymbol{W}_1^{(l)}\boldsymbol{a}_1 + \boldsymbol{b}_1$          ▷ linear layers preserve input-output relationship
12:      $\boldsymbol{a}_1 \leftarrow \boldsymbol{a}_0 + \Delta\boldsymbol{a}$          ▷ first-order Taylor approximation of activations
13: **end for**
14: $\Delta\boldsymbol{z} \leftarrow \Delta\boldsymbol{W}^{(L)}\boldsymbol{a}_0 + \boldsymbol{W}_1^{(L)}\Delta\boldsymbol{a} + \Delta\boldsymbol{b}^{(L)}$
15: **return** $\frac{1}{2}||\Delta\boldsymbol{z}||^2$

---

### 3.1 LINEARIZED ACTIVATION FUNCTION

Given parameters $\boldsymbol{\theta}_0$ and $\boldsymbol{\theta}_1$ of two networks, we linearize each step of the computation with respect to its inputs. For an MLP that alternates between linear layers and non-linear activation functions, the linear transformations are unmodified (since they are already linear), while the activation functions are replaced with a first-order Taylor approximation around their current inputs. Let $(\boldsymbol{W}_i^{(l)}, \boldsymbol{b}_i^{(l)})$ be the weights and biases of layer $l$ in network $i$.

$$\boldsymbol{s}_0^{(l)} = \boldsymbol{W}_0^{(l)}\boldsymbol{a}_0^{(l-1)} + \boldsymbol{b}_0^{(l)} \tag{6}$$

$$\boldsymbol{a}_0^{(l)} = \phi(\boldsymbol{s}_0^{(l)}) \tag{7}$$

$$\boldsymbol{s}_1^{(l)} = \boldsymbol{W}_1^{(l)}\boldsymbol{a}_1^{(l-1)} + \boldsymbol{b}_1^{(l)} \tag{8}$$

$$\boldsymbol{a}_1^{(l)} = \phi(\boldsymbol{s}_0^{(l)}) + \phi'(\boldsymbol{s}_0^{(l)}) \odot (\boldsymbol{s}_1^{(l)} - \boldsymbol{s}_0^{(l)}) \tag{9}$$

where $\phi'$ is the derivative of the ReLU function defined as $\phi'(\boldsymbol{s}) = \mathbb{1}\{\boldsymbol{s} > 0\}$. We define some additional useful notation for differences between preactivations and activations.

$$\Delta\boldsymbol{s}^{(l)} = \boldsymbol{s}_1^{(l)} - \boldsymbol{s}_0^{(l)} = \Delta\boldsymbol{W}^{(l)}\boldsymbol{a}_0^{l-1} + \boldsymbol{W}_1^{(l)}\Delta\boldsymbol{a}^{(l-1)} + \Delta\boldsymbol{b}^{(l)} \tag{10}$$

$$\Delta\boldsymbol{a}^{(l)} = \boldsymbol{a}_1^{(l)} - \boldsymbol{a}_0^{(l)} = \phi'(\boldsymbol{s}_0^{(l)}) \odot \Delta\boldsymbol{s}^{(l)} \tag{11}$$

with $\Delta\boldsymbol{W}^{(l)} = \boldsymbol{W}_1^{(l)} - \boldsymbol{W}_0^{(l)}$ and base cases $\boldsymbol{a}_0^{(0)} = \mathbf{x}$ and $\Delta\boldsymbol{a}^{(0)} = 0$. Observe that with $\boldsymbol{W}_0$ held fixed, the model parametrized using $\boldsymbol{W}_1$ is a linear network, i.e. the network's exact outputs are a linear function of its inputs. There are two significant differences between our approximation and parameter space linearization, which confer an advantage to the former. First, our linearization is with respect to inputs instead of parameters (Lee et al., 2019), hence capturing nonlinear interactions between the parameters in different layers. Second, the only computations that introduce linearization errors into our approximation are those involving activation functions, in contrast with other methods which suffer linearization errors for each layer, including linear layers, where nonlinear parameter dependencies exist. We note that linear networks are commonly used to model nonlinear training dynamics of neural networks (Saxe et al., 2013), hence we regard our weaker form of linearity as a significant advantage over Taylor approximations in parameter space. In this form, the linearized activation function trick applies to any linear (including fully-connected and convolutional) network with nonlinear activations.

### 3.2 BERNOULLI GATING

In the specific case of ReLU networks, our approximation only depends on the training data through the signs of all the preactivations, since passing layer inputs through $\phi$ and multiplying by the derivative $\phi'$, both result in a multiplication by 0 or 1. Hence, the maximum likelihood estimate (MLE) of

this multiplication factor corresponds to the MLE of a Bernoulli random variable and can be modeled as such. To achieve a compact approximation, we implement stochastic gating by replacing the activation signs with independent Bernoulli random variables for each unit and making a Gaussian approximation to the input distribution. We denote the vector of Bernoulli random variables for layer $l$ as $\mathbf{m}^{(l)}$, which are summarized by their corresponding mean parameters, $\mu^{(l)}$. Now we can rewrite equations 7 and 11 as $\boldsymbol{a}_0^{(l)} = \mathbf{m}^{(l)} \odot \boldsymbol{s}_0^{(l)}$ and $\Delta\boldsymbol{a}^{(l)} = \mathbf{m}^{(l)} \odot \Delta\boldsymbol{s}^{(l)}$, respectively, where $\odot$ denotes element-wise multiplication. Since FSD is defined as an expectation over the training distribution, we are interested in the moments of $\boldsymbol{a}$ and $\Delta\boldsymbol{a}$, which in turn depend on the first two moments of the input data and means $\mu$. We estimate $\mu$ with the maximum likelihood estimate (MLE) of $\mathbf{m}$, which is the average number of times a given unit is on during training. This can be computed efficiently in the forward pass as a simple moving average during the last epoch of training on a given dataset. The data moments are then propagated through the network, and multiplied by Bernoullis as required, finally resulting in the FSD term in Equation 5. The steps for this computation are detailed in the following Section 3.3 and Algorithm 1.

### 3.3 BGLN-D AND BGLN-S

We observe from equation 5 that the expectation of FSD can be written in terms of the first two moments of $\Delta\boldsymbol{z}$, or equivalently $\Delta\boldsymbol{s}^{(L)}$. Expectations and covariances of the outputs of each layer can be computed in terms of the expectations and covariances of the outputs of the previous layer. We propose computing these terms recursively by propagating the first two moments of $\boldsymbol{a}_0^{(l)}$ and $\Delta\boldsymbol{a}^{(l)}$ through the network. This can be done under our independence assumptions and using linearity of expectations. We thus obtain a deterministic estimate of the expected FSD, which we refer to as **BGLN-D**. The exact equations for this computation are shown in Appendix B. We can also compute a stochastic estimate by starting with the first two moments of the inputs, sampling $\boldsymbol{a}^{(0)}$ from a multivariate Guassian, and sampling Bernoulli random variables at every ReLU layer. This method, **BGLN-S**, is shown in Algorithm 1. Note that this is an unbiased estimate of the expected FSD. While we explicitly derive it for MLPs here, this estimate is not specific only to MLPs. It can be easily generalized to convolutional networks where the same random variables are sampled as required in the network. We use and evaluate this generalization for a continual learning task involving CIFAR-100, where the commonly used model architecture includes convolutions. We describe the corresponding computations in detail in Appendix C.

Note that storing moments of the data has less memory cost than storing sufficient subsets of the data itself. We present the above algorithms using the mean and covariance of the data. However, we point out that the data covariance can be replaced by its diagonal approximation, i.e. the variance of each dimension of the inputs. This imposes an independence assumption on the dimensions of the input data, analogous to the independence of Bernoulli random variables representing activations for each layer. This further reduces the memory cost of storing moments to the equivalent of two data points per dataset. We empirically investigate the effect of this approximation on continual learning benchmarks.

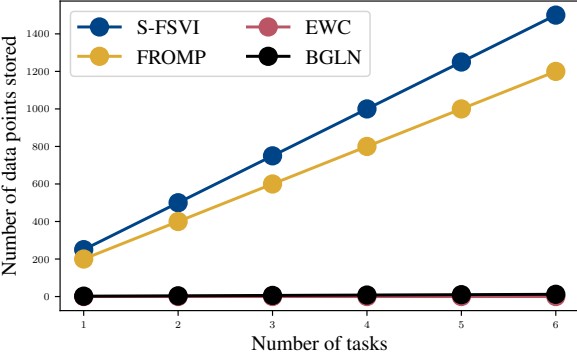

**Figure 2:** Comparison of number of datapoints stored for Split CIFAR100 as number of tasks increases. EWC stores none and BGLN stores the equivalent of two datapoints in the form of their moments, which is far lower than state-of-the-art nonparametric methods.

### 3.4 CLASS-CONDITIONAL ESTIMATES

It is also possible to a extend our method to a more fine-grained, class-conditioned approximation. The lower memory cost of our method allows for this. In particular, we can fit a mixture model, with one component per class. In this case, each class has its own associated data moments and mean parameters for the Bernoulli random variables. As expected, this results in improved performance in

our continual learning experiments, as shown by the results for BGLN-D (CW) and BGLN-S (CW) in Tables 1 and 2.

## 4 RELATED WORKS

Measuring distances between network parameters has been studied well in the existing literature. Here, we categorize the different approaches that are relevant to our proposed method.

**Network distance estimation.** Several works (Benjamin et al., 2018; Bernstein et al., 2020; Bae et al., 2022b) have highlighted the importance of measuring meaningful distances between neural networks. Benjamin et al. (2018) contrast training dynamics in parameter space and function space and observe that function space distances are often more useful than, and not always correlated with, weight space distances. Bae et al. (2022b) propose an amortized proximal optimization scheme (APO) that regularizes an FSD estimate to the previous iterate for second-order optimization. Natural gradient descent (Amari et al., 1995; Amari, 1998) can be interpreted as a steepest descent method, using a second-order Taylor approximation to the FSD (Pascanu & Bengio, 2014).

There are parametric approximations (Kirkpatrick et al., 2017; Ritter et al., 2018) of the FSD, which only utilize the parameters (and potentially related information) of the two networks in question. Nonparametric approximations typically store or optimize a small set of training inputs. For instance, Hessian-free optimization (Martens et al., 2010) and APO (Bae et al., 2022b) can be interpreted as using batches of data to approximate the FSD. Several non-parametric methods (Pan et al., 2020; Titsias et al., 2019; Rudner et al., 2022; Kapoor et al., 2021) for continual learning have been shown to give generally better empirical performance than parametric counterparts.

**Continual Learning.** Parisi et al. (2019); De Lange et al. (2021); Ramasesh et al. (2020); Normandin et al. (2021) have reviewed and surveyed the challenge of catastrophic forgetting in continual learning, along with benchmarks and metrics to evaluate different methods. Parametric methods focus on different approximations to the weight space metric matrix discussed above, like diagonal (Kirkpatrick et al., 2017, EWC), or Kronecker factorized (Ritter et al., 2018, OSLA) ones. Several methods are motivated as approximations to a posterior Gaussian distribution in a Bayesian setting (Ebrahimi et al., 2019), for instance through a variational lower bound (Nguyen et al., 2017) or via Gaussian process inducing points (Kapoor et al., 2021). Non-parametric methods (Kapoor et al., 2021; Titsias et al., 2019; Pan et al., 2020; Rudner et al.; 2022; Kirichenko et al., 2021) usually employ some form of experience replay of stored data points. Some of these methods (Pan et al., 2020) can be related to the Neural Tangent Kernel (Jacot et al., 2018, NTK), or in other words, network linearization. Doan et al. (2021) directly study forgetting in continual learning in the infinite width NTK regime. Mirzadeh et al. (2022) further study the impact of network widths on forgetting.

## 5 EXPERIMENTS

We evaluate the benefits of our method in practice, investigating the following questions: (1) What advantages do we gain by using our method in continual learning settings, relative to existing parametric and nonparametric FSD estimation techniques? (2) How well does our approximation estimate and minimize the true empirical FSD? (3) Can we use our FSD approximation to accurately compute influence functions without iterating through the full training data? In the results below, we find that (1) our methods are competitive with prior continual learning approaches despite having lower memory requirements than them, (2) it closely matches the true empirical FSD and (3) it gives a high correlation with influence functions using only the training data moments. Influence estimates can also be applied to identify mislabeled examples, which our method is able to do as well, as detailed in Appendix F.

### 5.1 CONTINUAL LEARNING

Following the formulation described in equation 2, we empirically evaluate BGLN and its variants.

**Datasets and architectures.** We visualize our method's performance on one-dimensional regression with two sequential tasks shown in Figure 1. More realistically, we test our method on image classification benchmarks used in prior works (Pan et al., 2020; Rudner et al., 2022), including Split

**Table 1:** Average classification accuracy across tasks on continual learning benchmarks and their associated memory costs. Methods are categorized as parametric, nonparametric, our BGLN methods, and ablations.

| Method | Split MNIST | Permuted MNIST | Split CIFAR100 | Memory Cost |
|---|---|---|---|---|
| EWC | 63.10 | 84.00 | $71.60 \pm 0.40$ | $2P$ |
| OSLA | 80.56 | 95.73 | 72.61 | $P + \sum_{l=1}^{L} p_l^2$ |
| VCL | 97.00 | $87.50 \pm 0.61$ | – | $2P$ |
| VCL (coreset) | 98.40 | 95.50 | $67.40 \pm 0.60$ | $2P + Nd$ |
| VAR-GP | $90.57 \pm 1.06$ | $\mathbf{97.20 \pm 0.08}$ | – | $2P + Nd + C^2N^2$ |
| FROMP | $99 \pm 0.04$ | $94.90 \pm 0.04$ | $76.20 \pm 0.20$ | $2P + Nd + C^2N^2$ |
| S-FSVI | $99.54 \pm 0.04$ | $95.76 \pm 0.02$ | $77.60 \pm 0.20$ | $2P + Nd + C^2N^2$ |
| NTK (coreset) | $99.50 \pm 0.09$ | $96.46 \pm 0.11$ | $\mathbf{78.23 \pm 0.20}$ | $2P + Nd$ |
| BGLN-S | $99.64 \pm 0.04$ | $96.36 \pm 0.12$ | $73.98 \pm 0.37$ | $P + A + d + d^2$ |
| BGLN-D | $99.72 \pm 0.03$ | $96.03 \pm 0.20$ | – | $P + A + d + d^2$ |
| BGLN-S (CW) | $99.77 \pm 0.05$ | $\mathbf{96.99 \pm 0.07}$ | $74.14 \pm 0.57$ | $P + C(A + d + d^2)$ |
| BGLN-D (CW) | $\mathbf{99.78 \pm 0.02}$ | $96.85 \pm 0.02$ | – | $P + C(A + d + d^2)$ |
| BGLN-D-Var | $99.64 \pm 0.04$ | $94.98 \pm 0.18$ | – | $P + A + 2d$ |
| BGLN-S-Var | $99.50 \pm 0.03$ | $96.36 \pm 0.13$ | $72.18 \pm 0.26$ | $P + A + 2d$ |
| BGLN-S (coreset) | $99.50 \pm 0.03$ | $96.36 \pm 0.13$ | $72.89 \pm 0.4$ | $P + A + Nd$ |

**Table 2:** Backward transfer on continual learning benchmarks (higher is better).

| Method | Split MNIST | Permuted MNIST | Split CIFAR100 |
|---|---|---|---|
| FROMP | $-0.50 \pm 0.20$ | $-1.00 \pm 0.10$ | $-2.60 \pm 0.90$ |
| S-FSVI | $-0.21 \pm 0.06$ | $-0.65 \pm 0.21$ | $\mathbf{-2.50 \pm 0.20}$ |
| BGLN-S | $\mathbf{-0.04 \pm 0.03}$ | $-0.41 \pm 0.08$ | $-7.33 \pm 0.27$ |
| BGLN-D | $-0.09 \pm 0.04$ | $-0.56 \pm 0.04$ | – |
| BGLN-S (CW) | $-0.18 \pm 0.06$ | $\mathbf{-0.37 \pm 0.14}$ | $-6.13 \pm 0.44$ |
| BGLN-D (CW) | $-0.07 \pm 0.07$ | $-1.17 \pm 0.07$ | – |

MNIST, Permuted MNIST and Split CIFAR100, with standard architectures for fair comparison. More details about the datasets, architectures and hyperparameters can be found in Appendix E.2.

**Evaluation Metrics.** We evaluate the success of different methods based on average final accuracy across tasks and the backward transfer metric (Lopez-Paz & Ranzato, 2017), defined as the average increase in task accuracy after training on all tasks relative to after training on that task.

**Results.** Figure 1 shows the functions learned by different methods when sequentially trained on two one-dimensional regression tasks. BGLN retains good predictions on both tasks, while EWC and exact NTK suffer catastrophic forgetting. We hypothesize that this difference in performance is due to important nonlinearities between network parameters that EWC and NTK approximations are unable to capture.

We compare our method's performance on the above evaluation metrics on image classification benchmarks to existing parametric and nonparametric methods in Tables 1 and 2. BGLN methods outperform other parametric methods (EWC, OSLA and VCL) across datasets and are competitive with the state-of-the-art nonparametric methods. Further, using class-conditional approximations improves results. On Split CIFAR100, BGLN methods start to bridge the gap

**Table 3:** Memory cost notation.

| | |
|---|---|
| $p_l$ | # parameters in layer $l$ |
| $P$ | # parameters $= \sum_{l=1}^{L} p_l$ |
| $A$ | # activations $< P$ |
| $d$ | data dimension |
| $N$ | coreset size |
| $C$ | # classes |

between parametric and nonparametric methods, while the NTK approximation, which uses a coreset, outperforms all previous works. We hypothesize that methods like FROMP can be viewed as approximations to the NTK method and our current result indicates that directly estimating FSD via NTK can improve continual learning performance. Table 2 compares the backward transfer metric for our methods against the current state-of-the-art in continual learning. Both FROMP (Pan et al., 2020) and S-FSVI (Rudner et al., 2022) store coresets of data, yet BGLN outperforms them on the MNIST tasks, while lagging behind on Split CIFAR100. We show how the memory requirement scales in terms of number of datapoints stored in Figure 2 and analytically compare this cost for

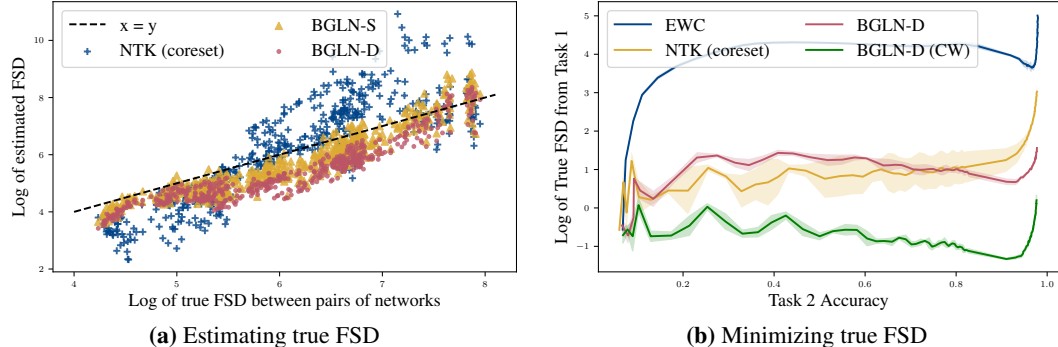

**Figure 3:** Comparison of different FSD estimators for networks trained on Permuted MNIST. Note that FSD values are plotted on log-scale. **(a)** Compared to NTK, BGLN consistently gives closer FSD values to the true empirical FSD. **(b)** While training on task 2, FSD from the optimal task 1 parameters increases with task 2 accuracy. Optimizing BGLN-D and class-conditioned BGLN-D (CW) effectively minimizes the true FSD.

different methods in Table 1, using notation in Table 3. This makes the memory advantage of our methods explicit. Further details on computing these costs are in Appendix E.1.

## 5.2 COMPARING FSD ESTIMATORS

We empirically assess how well our method approximates and minimizes the true empirical FSD between two sets of parameters. Motivated by the continual learning application, we consider the Permuted MNIST task and its sequentially learned network parameters. As seen above, training on new tasks risks catastrophic forgetting of predictions on old data and can significantly change the learned function space. We evaluate different FSD estimators by measuring (1) how close the estimated FSD is to the true empirical FSD computed over the full dataset and (2) how much the true empirical FSD is minimized when the estimated FSD is optimized during training. We draw attention to the NTK estimator which exactly linearizes the network using a coreset of datapoints. Figure 3a plots the estimated FSD versus the true empirical FSD between several pairs of networks trained on different tasks, for different number of iterations and with different learning rates. Both our methods, BGLN-S and BGLN-D consistently have closer estimates to the true empirical FSD than NTK across the range of networks considered. As shown in Figure 3b, our methods also minimize the true empirical FSD more effectively than EWC and NTK.

## 5.3 INFLUENCE FUNCTION ESTIMATION

To demonstrate that BGLN can successfully be applied to other applications involving FSD approximation, we consider an influence function estimation task (Cook, 1979; Hampel, 1974; Koh & Liang, 2017). Given parameters $\boldsymbol{\theta}_0$ trained on some dataset $\mathcal{D}_{\text{train}}$ that is drawn from a distribution $p_{\text{train}}$, influence functions approximate the parameters $\boldsymbol{\theta}_-$, obtained by training without a particular point $(\mathbf{x}, \mathbf{y}) \in \mathcal{D}_{\text{train}}$. The difference in loss between $\boldsymbol{\theta}_0$ and $\boldsymbol{\theta}_-$ is an indicator for how much influence $(\mathbf{x}, \mathbf{y})$ has on the trained network.

Given a data point, $(\boldsymbol{x}, \boldsymbol{y})$, that we are interested in removing, Bae et al. (2022a) recently showed how to express influence functions in neural networks as optimizing the following proximal Bregman response function (PBRF):

$$\boldsymbol{\theta}_- = \arg\min_{\boldsymbol{\theta} \in \mathbb{R}^d} D_B(\boldsymbol{\theta}, \boldsymbol{\theta}_0, p_{\text{train}}) + \frac{\lambda}{2} \|\boldsymbol{\theta} - \boldsymbol{\theta}_0\|^2 - \frac{1}{N} \mathcal{L}(f(\boldsymbol{x}, \boldsymbol{\theta}), \boldsymbol{y}), \tag{12}$$

where $N$ is the total number of training datapoints. Here, $D_B$ is the Bregman divergence defined on network outputs and measures the FSD between $\boldsymbol{\theta}$ and $\boldsymbol{\theta}_0$ for training distribution $p_{\text{train}}$, as defined in equation 1 from Section 2. For common loss functions such as squared error and cross-entropy, the Bregman divergence term is equivalent to the training error on a dataset where the original targets are changed with soft targets produced by $\boldsymbol{\theta}_0$. The second term is a proximity term with strength $\lambda > 0$, whose role is to prevent the parameters from moving too far in parameter space. Finally, the last term maximizes the loss of the data point we are interested in removing. Intuitively, the PBRF maximizes the loss of data we would like to remove while constraining the network in both function and parameter space so that the predictions and losses of other training examples are not affected.

Similar to the issues encountered in continual learning, the computation of the FSD term requires storing and accessing the entire training data even after the original network is trained. In practice, Bae et al. (2022a) iteratively sample batches of the training data to compute and minimize the FSD term. However, training data may not be accessible and incurs a large memory cost to store. We can employ our parametric method to estimate the FSD in PBRF without accessing the training data, but only its first two moments.

**Datasets and architectures.** We first train a MLP with 2 hidden layers and ReLU activation for 200 epochs on regression datasets from the UCI benchmark (Dua & Graff, 2017). Then, we randomly select 50 independent data points to be removed. Using the PBRF objective, we compute the difference in loss after removing a single data point, which is commonly referred to as the self-influence score (Koh & Liang, 2017; Schioppa et al., 2022). Then, we apply BGLN-D to approximate the FSD term in the PBRF objective and compare it with other approximations such as EWC and Conjugate Gradient (CG) (Martens et al., 2010).

**Evaluation Metric and Results.** Since the PBRF can be considered the ground truth for influence estimation, we compare the correlation of EWC, CG, and BGLN with the PBRF. We used both Pearson correlation (Sedgwick, 2012) and Spearman rank-order correlation (Spearman, 1961) to measure this alignment. We show the results in Table 4. Across all datasets, without having to save or iterate over the full training dataset, BGLN-D achieves a higher correlation with the PRBF objective compared to EWC and CG.

**Table 4:** Comparison of train loss differences computed by EWC, conjugate gradient (CG), and FSD. We show Pearson (P) and Spearman rank-order (S) correlation with the PBRF estimates.

| Dataset | EWC | | CG | | BGLN-D | |
|---|---|---|---|---|---|---|
| | P | S | P | S | P | S |
| Concrete | 0.78 | 0.57 | 0.92 | 0.94 | **0.96** | **0.97** |
| Energy | 0.68 | 0.39 | 0.97 | 0.98 | **0.99** | **0.98** |
| Housing | 0.86 | 0.33 | 0.92 | **0.89** | **0.95** | 0.83 |
| Kinetics | 0.36 | 0.30 | 0.88 | 0.86 | **0.99** | **0.99** |
| Wine | 0.97 | 0.70 | 0.99 | **0.94** | **0.99** | 0.90 |

## 5.4 ABLATIONS

We analyze the impact of the different choices our method makes by ablating them one by one. (1) **BGLN-D-Var** and **BGLN-S-Var** use a diagonal approximation to the data covariance which effectively assumes independence between features. We observe in Table 1 that this approximation does not harm performance significantly. (2) **BGLN-S (coreset)** teases apart the effect of sampling data using its first two moments by running the stochastic version of our method with a small number of actual data samples. Results on backward transfer for the same can be found in Appendix E.4.

## 6 CONCLUSIONS

We addressed the problem of compactly summarizing a model's predictions on a given dataset, and formulated it as approximating neural network FSD. We found that BGLN methods, our novel parametric approximations to FSD, capture nonlinearities between network parameters and are much more memory-efficient than prior works. We demonstrated the closeness of our estimate to the true FSD as well as its application in two use-cases. In continual learning, BGLN methods outcompete existing methods on common benchmarks without storing any data samples. Furthermore, they effectively estimate influence functions without iterating over or storing the full dataset. We note that while our methods make a closer approximation to the true FSD in principle, they are limited by independence and probabilistic modeling assumptions. Empirically, these may explain its lower but competitive performance on Split CIFAR100. Interestingly, the direct NTK approximation outperforms all previous methods, which makes optimizing its memory-efficiency an attractive direction for future work. Further, extending the formulation of FSD approximation to other applications like model editing or unlearning are exciting research avenues. We hope that our work inspires methods to further bridge the gap to nonparametric methods' performance in a memory-efficient manner.

## 7 REPRODUCIBILITY STATEMENT

All details about datasets, model architectures and training can be found in the Appendix. We also plan to release code that reproduces the results in this paper once it is public.

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

APPENDIX

## A  NOTATION

| | |
|---|---|
| $f$ | Function corresponding to a neural network |
| $\phi$ | ReLU activation function |
| $\phi'$ | Derivative of a function $\phi$ |
| $\boldsymbol{s}_l$ | Preactivations at layer $l$ |
| $\boldsymbol{a}_l = \phi(\boldsymbol{s}_l)$ | Activations at layer $l$ |
| $\mathbf{x}$ | Input data |
| $\boldsymbol{y}$ | Target data |
| $\mathbf{x}^{(i)}$ | Input data point $i$ |
| $p_{\text{data}}$ | Data distribution from which $\mathbf{x}$ is sampled |
| $d$ | Data dimension |
| $N$ | Number of data points in a coreset |
| $L$ | Number of layers in a network |
| $C$ | Number of classes in a classification task |
| $T$ | Number of tasks |
| $\boldsymbol{\theta}$ | Parameters of a neural network |
| $\boldsymbol{\theta}_t$ | Parameters obtained after training on task $t$ |
| $p_l$ | Number of parameters in layer $l$ |
| $P$ | Total number of parameters in a network $= \sum_{l=1}^{L} p_l$ |
| $\boldsymbol{z} = f(\mathbf{x}; \boldsymbol{\theta})$ | Prediction of the network $f$ on $\mathbf{x}$ parameterized by $\boldsymbol{\theta}$ |
| $\rho$ | Output space distance, for instance Euclidean distance |
| $D(\boldsymbol{\theta}_0, \boldsymbol{\theta}_1)$ | Function space divergence between networks parameterized by $\boldsymbol{\theta}_0$ and $\boldsymbol{\theta}_1$ |
| $\boldsymbol{J} = \nabla_{\boldsymbol{\theta}} f$ | Network Jacobian |
| $\boldsymbol{G}_{\boldsymbol{\theta}}$ | Weight space metric matrix |
| $\boldsymbol{F}_{\boldsymbol{\theta}}$ | Fisher information matrix |
| $\mathbf{m}$ | Bernoulli random variable |
| $\mu$ | Bernoulli mean parameter |

## B  RECURSION EQUATIONS FOR BGLN-D

We derive the deterministic version of our algorithm by taking expectations and covariances for the quantities in equations 6 to 11 (rewritten using Bernoulli variables). We use linearity of expectations and our assumptions of independence between the Bernoulli random variables. We also assume $\text{Cov}(a_0, \Delta a)$ is close to 0 and ignore it in our computations. This assumption is tested empirically in our experiments and we find that it does not severely move the FSD estimate away from the true empirical FSD (see Figure 3a). The complete steps for BGLN-D computations are given in Algorithm 2.

---

**Algorithm 2** BGLN-D

---

**Require:** $\mathbb{E}[\mathbf{x}], \text{Cov}(\mathbf{x}), \{\boldsymbol{W}, \boldsymbol{b}\}_1^{L-1}, \{\mu\}_1^{L-1}$

1: $\mathbb{E}[\boldsymbol{a}_0], \mathbb{E}[\boldsymbol{a}_1] \leftarrow \mathbb{E}[\mathbf{x}]$
2: $\text{Cov}(\boldsymbol{a}_0), \text{Cov}(\boldsymbol{a}_1) \leftarrow \text{Cov}(\mathbf{x})$
3: $\mathbb{E}[\Delta a], \text{Cov}(\Delta a) \leftarrow 0$
4: **for** $l \leftarrow 1$ to $L-1$ **do**
5:     $\mathbb{E}[\boldsymbol{s}_0] \leftarrow \boldsymbol{W}_0^{(l)} \mathbb{E}[\boldsymbol{a}_0] + \boldsymbol{b}_0^{(l)}$
6:     $\mathbb{E}[\Delta s] \leftarrow \Delta \boldsymbol{W}^{(l)} \mathbb{E}[\boldsymbol{a}_0] + \boldsymbol{W}_1^{(l)} \mathbb{E}[\Delta a] + \Delta \boldsymbol{b}^{(l)}$
7:     $\mathbb{E}[\boldsymbol{a}_0] \leftarrow \mu^{(l)} \odot \mathbb{E}[\boldsymbol{s}_0]$
8:     $\mathbb{E}[\Delta a] \leftarrow \mu^{(l)} \odot \mathbb{E}[\Delta s]$
9:     $\text{Cov}(\boldsymbol{s}_0) \leftarrow \boldsymbol{W}_0^{(l)} \text{Cov}(\boldsymbol{a}_0) \boldsymbol{W}_0^{(l)T}$
10:     $\text{Cov}(\Delta s) \leftarrow \Delta \boldsymbol{W}^{(l)} \text{Cov}(\boldsymbol{a}_0) \Delta \boldsymbol{W}^{(l)T} + \boldsymbol{W}_1^{(l)} \text{Cov}(\Delta a) \boldsymbol{W}_1^{(l)T}$
11:     $\text{Cov}(\boldsymbol{a}_0) \leftarrow (\mu^{(l)} \mu^{(l)T}) \odot \text{Cov}(\boldsymbol{s}_0)$
12:     $\text{Cov}(\Delta a) \leftarrow (\mu^{(l)} \mu^{(l)T}) \odot \text{Cov}(\Delta s)$
13: **end for**
14: $\mathbb{E}[\Delta z] \leftarrow \Delta \boldsymbol{W}^{(L)} \mathbb{E}[\boldsymbol{a}_0] + \boldsymbol{W}_1^{(L)} \mathbb{E}[\Delta a] + \Delta \boldsymbol{b}^{(L)}$
15: $\text{Cov}(\Delta z) \leftarrow \Delta \boldsymbol{W}^{(L)} \text{Cov}(\boldsymbol{a}_0) \Delta \boldsymbol{W}^{(L)T} + \boldsymbol{W}_1^{(L)} \text{Cov}(\Delta a) \boldsymbol{W}_1^{(L)T}$
16: **return** $\frac{1}{2} ||\mathbb{E}[\Delta z]||^2 + \frac{1}{2} tr \, \text{Cov}(\Delta z)$

---

**Algorithm 3** BGLN-S (Conv)

---

**Require:** $\mathbb{E}[\mathbf{x}], \text{Cov}(\mathbf{x}), \{\texttt{layer}\}_1^{L-1}, \{\mu\}_1^{L-1}$

1: $\mathbb{E}[\boldsymbol{a}_0], \mathbb{E}[\boldsymbol{a}_1] \leftarrow \mathbb{E}[\mathbf{x}]$
2: $\text{Cov}(\boldsymbol{a}_0), \text{Cov}(\boldsymbol{a}_1) \leftarrow \text{Cov}(\mathbf{x})$
3: $\boldsymbol{a}_0, \boldsymbol{a}_1 \sim \mathcal{N}(\mathbb{E}[\boldsymbol{a}_0], \text{Cov}(\boldsymbol{a}_0))$
4: $\Delta a \leftarrow 0$
5: **for** $l \leftarrow 1$ to $L-1$ **do**
6:     **if** $\texttt{layer}$ is $\texttt{Conv}$ or $\texttt{FC}$ **then**
7:         $\boldsymbol{s}_0 \leftarrow \texttt{layer}(\boldsymbol{a}_0, \texttt{grad=False})$
8:         $\boldsymbol{s}_1 \leftarrow \texttt{layer}(\boldsymbol{a}_1)$
9:     **else if** $\texttt{layer}$ is $\texttt{ReLU}$ **then**
10:         $\Delta \boldsymbol{s} = \boldsymbol{s}_1 - \boldsymbol{s}_0$
11:         $\mathbf{m} \sim Ber(\mu^{(l)})$
12:         $\Delta \boldsymbol{a} \leftarrow \mathbf{m} \odot \Delta \boldsymbol{s}$
13:     **else if** $\texttt{layer}$ is $\texttt{Dropout}$ **then**
14:         pass
15:     **else**
16:         $\boldsymbol{a}_0 \leftarrow \texttt{layer}(\boldsymbol{a}_0)$
17:         $\boldsymbol{a}_1 \leftarrow \texttt{layer}(\boldsymbol{a}_1)$
18:     **end if**
19: **end for**
20: $\Delta \boldsymbol{z} \leftarrow \boldsymbol{s}_1 - \boldsymbol{s}_0$
21: **return** $\frac{1}{2} ||\Delta \boldsymbol{z}||^2$

---

## C  GENERALIZATION TO CONVOLUTIONAL NETWORKS

The generalization of BGLN-S to convolutional networks involves passing the inputs sampled using the data moments through the network. In convolutional networks, ReLU activation is usually applied after the convolutional and the fully connected layers. At each ReLU, we use the Bernoulli mean parameters to sample activation signs and obtain the difference of activations, $\Delta a$. We treat any dropout layers as they are treated at evaluation time, i.e. as the identity function. Finally, the Euclidean distance between the final layer outputs leads to the stochastic estimate of the FSD. The complete procedure is shown in Algorithm 3.

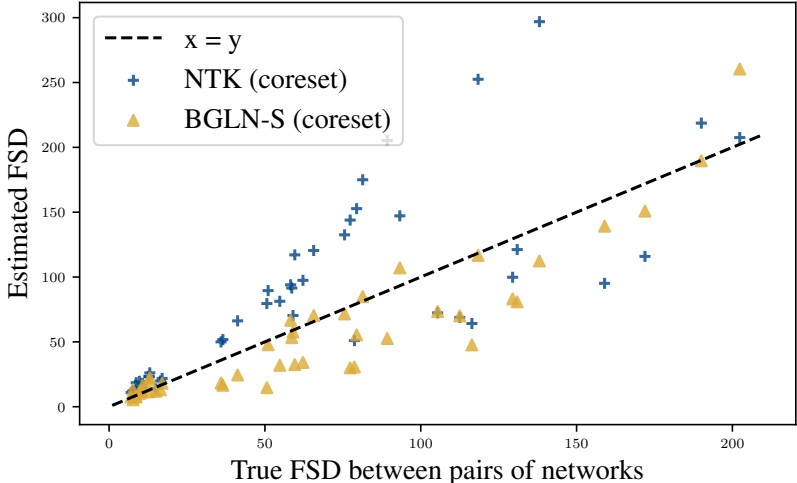

**Figure 4:** Estimating true FSD with BGLN-S (coreset) and NTK (coreset) on networks trained using Split CIFAR100.

## D  COMPARING FSD ESTIMATORS

To directly compare the NTK approximation with the linearized activation function trick, we compute the estimated FSD between pairs of networks using these two kinds of linearization, when provided with the same information, and plot them against the true emprical FSD. Hence, both estimators are provided with the same coreset of datapoints. In the case of BGLN-S (coreset), activations are computed using this coreset directly instead of Bernoulli sampling. Figure 4 visualizes this comparison for networks trained using tasks in Split CIFAR100. BGLN-S estimates correlate better with the true FSD than NTK, hence corroborating our intuition about linearizing activation functions. To quantify this difference, we also measure the Spearman rank-order and Kendall's Tau correlation coefficients for each estimator with the true FSD. BGLN-S obtains values 96.36 and 79.19, respectively, outperforming NTK, which obtains 86.42 and 71.71, respectively.

## E  CONTINUAL LEARNING

### E.1  MEMORY COST ANALYSIS

We follow the notation in Table 3 to denote $p_l$ as the number of parameters in layer $l$, $P = \sum_{l=1}^{L} p_l$ as the total number of parameters in the network, $A$ as the number of activations in the network (note that $A < P$), $d$ as the data dimension, $N$ as the number of samples in a coreset, and $C$ as the number of classes in the continual learning classification setting. We can now write analytic expressions for the memory cost incurred by the different methods considered in the continual learning experiments, as shown in Table 1. Below we arrive at these expressions for each task that the model is continually trained on.

- **EWC:** EWC requires storing one value for each parameter of the network and one value for each diagonal element of the $P \times P$ Fisher information matrix, resulting in a cost of $2P$.

- **OSLA:** OSLA approximates the Fisher information matrix as a block diagonal matrix, storing $p_l^2$ elements for each block corresponding to layer $l$. This gives a cost of $P + \sum_{l=1}^{L} p_l^2$.

- **VCL:** VCL stores two pieces of information for the variational distribution for each parameter, one for the mean and one for the variance in the diagonal approximation, incurring a cost equal to $2P$.

- **VCL (coreset):** The "coreset" variant of VCL additionally stores $N$ datapoints, increasing the memory cost by $Nd$.

- **VAR-GP, FROMP, S-FSVI:** These methods all store two values per parameter, similar to VCL. Further, they require a coreset of datapoints and/or inducing points, as well as a $NC \times NC$ kernel matrix.
- **NTK (coreset):** The NTK approximation stores one value for each parameter of the network and one for the $P-$dimensional Jacobian-vector product used to linearize the network. It further requires a coreset of $N$ datapoints, result in a $2P + Nd$ cost.
- **BGLN-S, BGLN-D:** Our methods store each parameter once, a Bernoulli mean value for each activation and the first two moments of the data, incurring a cost of $P + A + d + d^2$.
- **BGLN-S (CW), BGLN-D (CW):** The classwise variants of our methods store separate Bernoulli means and data moments for each class, which scales those corresponding memory costs by C, i.e., $P + C(A + d + d^2)$.
- **BGLN-S (Var), BGLN-D (Var):** The "Var" variants of our methods make a diagonal approximation to the data covariance (second moment), hence storing only $d$ values for it. This further reduces memory cost to $P + A2d$.
- **BGLN-S (coreset):** The "coreset"" ablation of BGLN-S requires storing $N$ datapoints instead of the data moments, giving a $P + A + Nd$ cost.

## E.2 EXPERIMENTAL DETAILS

**Datasets.** Split MNIST consists of five binary prediction tasks to classify non-overlapping pairs of MNIST digits (Deng, 2012). Permuted MNIST is a sequence of ten tasks to classify ten digits, with a different fixed random permutation applied to the pixels of all training images for each task. Finally, Split CIFAR100 consists of six ten-way classification tasks, with the first being CIFAR10 (Krizhevsky et al., a), and subsequent ones containing ten non-overlapping classes each from the CIFAR100 dataset (Krizhevsky et al., b).

**Architectures.** We use standard architectures used by existing methods for fair comparison. For regression and the MNIST experiments, we use a MLP with two fully connected layers and ReLU activation. For Split CIFAR100, we use a network with four convolutional layers, followed by two fully connected layers, and ReLU activation after each. Both Split MNIST and Split CIFAR100 models have a multiheaded final layer.

**Hyperparameters.** We have performed a grid search over some key hyperparameters and used the ones that resulted in the best final average accuracy across all tasks. All hyperparameter search was done with random seed 42. We then took that best set of hyperparameters, repeated our experiments on seeds 20, 21, 22, and reported the average and standard deviation of our results.

For the learning rate, we used 0.001 for all CL experiments except the BGLN-S method for Split MNIST and BGLN-D method for Permuted MNIST, where we used 0.0001 instead.

We used the same number of epochs on each CL task and the exact numbers are reported in Table 5. On the first task, all MNIST experiments used the same number of epochs as the subsequent CL tasks while CIFAR100 experiments used 200 epochs on the first task.

To compute the Bernoulli mean parameters for our stochastic gating implementation, we used simple averaging as the default, but also explored exponential moving averaging. While there was not much difference in performance, we report the momentum values that reproduce our results. All MNIST experiments had a momentum value of $1/$batch_size. Note that this momentum value of $1/$batch_size corresponds to simple moving average. For CIFAR100 experiments, we used 0.99 for BGLN-S (CW) and NTK, and $1/$batch_size for BGLN-S.

For each method and dataset, the scaling factor for FSD penalty, $\lambda_{\text{FSD}}$, is reported in Table 6. Similarly, batch size is reported in Table 7.

**Evaluation Metrics.** In addition to average accuracy over tasks, we measure the backward transfer metric. For $T$ tasks, let $R_{i,j}$ be the classification accuracy on task $t_j$ after training on task $t_i$. Then, backward transfer is given by the following formula.

$$\frac{1}{T-1} \sum_{i=1}^{T-1} R_{T,i} - R_{i,i}$$

**Table 5:** CL tasks training epochs used in CL experiments.

| Method | Split MNIST | Permuted MNIST | Split CIFAR100 |
|---|---|---|---|
| **NTK (coreset)** | 15 | 5 | 80 |
| **BGLN-S** | 15 | 15 | 45 |
| **BGLN-D** | 15 | 15 | - |
| **BGLN-S (CW)** | 15 | 15 | 85 |
| **BGLN-D (CW)** | 15 | 15 | - |

**Table 6:** FSD scale used in CL experiments.

| Method | Split MNIST | Permuted MNIST | Split CIFAR100 |
|---|---|---|---|
| **NTK (coreset)** | 1 | 1 | 0.005 |
| **BGLN-S** | 5 | 1 | 0.0005 |
| **BGLN-D** | 0.1 | 0.005 | - |
| **BGLN-S (CW)** | 2 | 1 | 0.0000001 |
| **BGLN-D (CW)** | 0.1 | 0.005 | - |

**Table 7:** Batch size used in CL experiments.

| Method | Split MNIST | Permuted MNIST | Split CIFAR100 |
|---|---|---|---|
| **NTK (coreset)** | 256 | 256 | 512 |
| **BGLN-S** | 32 | 128 | 512 |
| **BGLN-D** | 32 | 128 | - |
| **BGLN-S (CW)** | 32 | 128 | 512 |
| **BGLN-D (CW)** | 32 | 128 | - |

**Table 8:** Ablation of backward transfer on continual learning benchmarks (higher is better).

| Method | Split MNIST | Permuted MNIST | Split CIFAR100 |
|---|---|---|---|
| **BGLN-D-Var** | $-0.18 \pm 0.09$ | $-3.90 \pm 0.49$ | - |
| **BGLN-S-Var** | $-0.13 \pm 0.11$ | $-0.41 \pm 0.08$ | $-9.25 \pm 0.06$ |
| **BGLN-S (coreset)** | $-0.13 \pm 0.11$ | $-0.41 \pm 0.08$ | $-7.50 \pm 0.51$ |

### E.3 TASK-WISE CLASSIFICATION RESULTS

We show in Figure 5 the task-wise accuracies on the Split CIFAR100 benchmark after training on all tasks is complete, drawing a comparison between our methods (BGLN-S and BGLN-S (CW)), NTK (with coreset) and a nonparametric state-of-the-art method, FROMP.

### E.4 ABLATION STUDY - BACKWARD TRANSFER

We include further results on the performance of our ablations on the backward transfer metric in Table 8.

## F INFLUENCE FUNCTION ESTIMATION

### F.1 EXPERIMENTAL DETAILS

We used Concrete, Energy, Housing, Kinetics, and Wine datasets from the UCI collection (Dua & Graff, 2017). For all datasets, we normalized the training dataset to have a mean of 0 and a standard deviation of 1. We used a 2-hidden layer MLP with 128 hidden units and the base network was

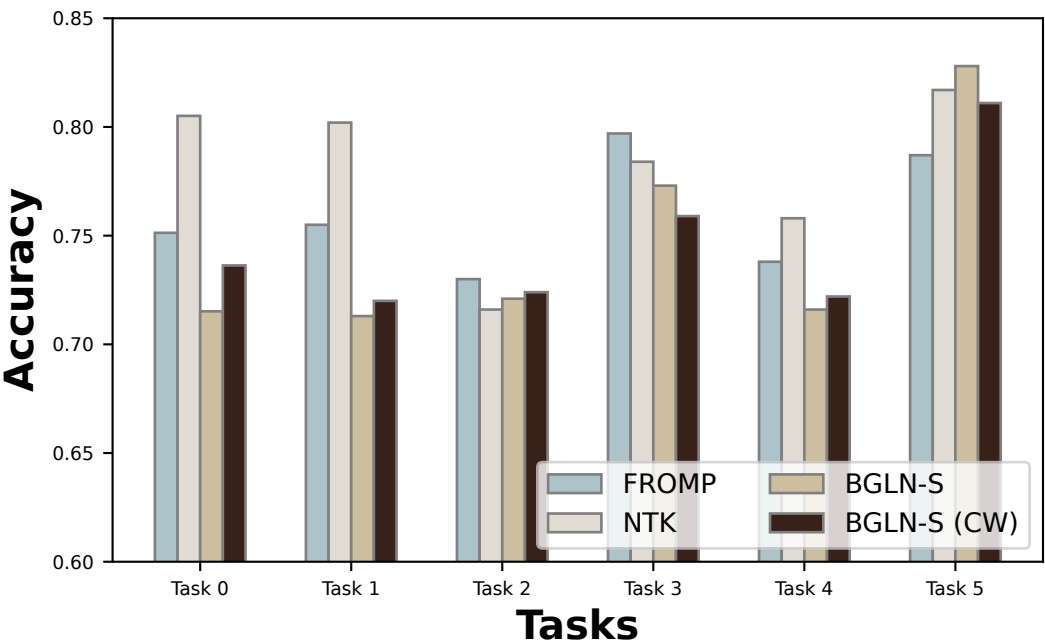

**Figure 5:** Comparison of task-wise accuracies on the Split CIFAR100 benchmark after training on all tasks is complete, for our methods, NTK and a nonparametric state-of-the-art method, FROMP.

trained for 200 epochs with SGD and a batch size of 128. We performed hyperparameter searches over the learning rate in the range {0.3, 0.1, 0.03, 0.01, 0.003, 0.001} and selected the learning rate based on the validation loss.

For each random data point selected, we optimized the PBRF objective for additional 20 epochs from the base network. The FSD term was computed stochastically with a batch size of 128. Similarly, both EWC and BGLN were trained with the same configuration but with the corresponding approximation to the FSD term.

### F.2 MISLABELED EXAMPLE IDENTIFICATION

A common application of the influence function is the detection of influential or mislabeled examples. Intuitively, if some fraction of the training data labels is corrupted, they would behave as outliers and have a greater influence on the training loss. One approach to efficiently detect and fix these examples is to prioritize and examine training inputs with higher self-influence scores.

We use 10% of the MNIST dataset and corrupt 10% of the data by assigning random labels to it. We train a two-layer MLP with 1024 hidden units and ReLU activation using SGD with a batch size of 128. Then, we use EWC and BFLN to approximate the FSD term in the PBRF objective in equation 12 for each data point. We also compare these methods against randomly sampling datapoints to check for corruptions. The results are summarized in Figure 6. Both PBRF and BGLN show significantly improved performance compared to the random baseline.

### G LINEARIZATION ERROR ANALYSIS OF BGLN-S

We can analyze the error due to linearization of BGLN-S as compared to the true FSD, propagated through each step of the network's computation. Following notation as in Section 3, let $\Delta z$ and $\Delta \hat{z}$ denote the true difference between network outputs and the difference between network outputs estimated after applying the linearized activation functions trick. We are interested in the error in this estimation,

$$err(\boldsymbol{z}) = \Delta \boldsymbol{z} - \Delta \hat{\boldsymbol{z}}. \tag{13}$$

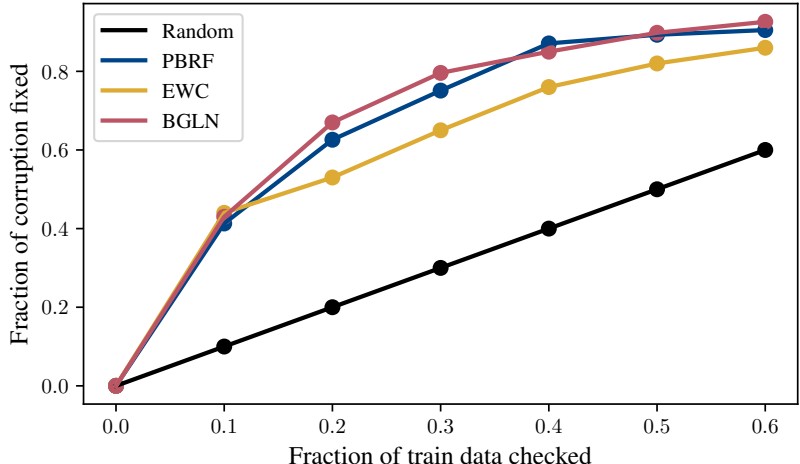

**Figure 6:** Effectiveness of BGLN in detecting mislabeled examples. BGLN can approximate the FSD term in the PBRF objective accurately and be used in applications involving influence functions without explicitly storing or iterating over the dataset.

As errors propagate through the network, we consider the error after each step of computation. Accordingly, we define,

$$err(\boldsymbol{s}^{(l)}) = \Delta \boldsymbol{s}^{(l)} - \Delta \hat{\boldsymbol{s}}^{(l)} \text{ and } err(\boldsymbol{a}^{(l)}) = \Delta \boldsymbol{a}^{(l)} - \Delta \hat{\boldsymbol{a}}^{(l)}. \tag{14}$$

The key observation here is that errors are only introduced due to the linearized activation computations. Also note that, given two networks, we linearize the activations of only one of them, leaving the other's computations accurate.

In the case of linear (fully-connected or convolutional) networks with continuously differentiable activation functions, the error introduced due to activations at any layer is the second-order Taylor error,

$$err(\boldsymbol{a}^{(l)}) = \mathcal{O}(||\boldsymbol{s}_1^{(l)} - \boldsymbol{s}_0^{(l)}||^2). \tag{15}$$

Any subsequent computation then has error only due to $err(\boldsymbol{a}^{(l)})$, with no further errors introduced due to linear layers. For instance, for fully-connected layers of the two networks parametrized using $\boldsymbol{W}_0^{(l)}$ and $\boldsymbol{W}_1^{(l)}$, which have as inputs $\boldsymbol{a}_0^{(l-1)}$ and $\boldsymbol{a}_1^{(l-1)}$, the estimation error is given by

$$err(\boldsymbol{s}^{(l)}) = \Delta \boldsymbol{W}^{(l)} \boldsymbol{a}_0^{l-1} + \boldsymbol{W}_1^{(l)} \Delta \boldsymbol{a}^{(l-1)} + \Delta \boldsymbol{b}^{(l)} \tag{16}$$

$$- \Delta \boldsymbol{W}^{(l)} \hat{\boldsymbol{a}}_0^{l-1} + \boldsymbol{W}_1^{(l)} \Delta \hat{\boldsymbol{a}}^{(l-1)} + \Delta \boldsymbol{b}^{(l)} \tag{17}$$

$$= \Delta \boldsymbol{W}^{(l)} (\boldsymbol{a}_0^{l-1} - \hat{\boldsymbol{a}}_0^{l-1}) + \boldsymbol{W}_1^{(l)} (\Delta \boldsymbol{a}^{(l-1)} - \Delta \hat{\boldsymbol{a}}^{(l-1)}) \tag{18}$$

$$= \boldsymbol{W}_1^{(l)} err(\boldsymbol{a}^{(l-1)}) \tag{19}$$

where the last line is given by the fact that $\boldsymbol{a}_0^{l-1} = \hat{\boldsymbol{a}}_0^{l-1}$ since activations $\boldsymbol{a}_0^{(l)}$ remain exact while we linearize computations for $\boldsymbol{a}_1^{(l)}$.

This error analysis formalizes the intuition that as errors are propagated through the network, they are only introduced due to linearized activation computations. The error incurred after a linear layer is a linear function of the error in its inputs, adding no new errors of its own.

