# OpenReview forum: "Efficient parametric approximations of neural net function space distance"
_ICLR.cc/2023/Conference — Submitted to ICLR 2023_

### Official Review · Reviewer_cRKz · 2022-10-25

**Confidence:** 2
**Correctness:** 4
**Technical Novelty And Significance:** 4
**Empirical Novelty And Significance:** 3
**Recommendation:** 8

**Clarity, Quality, Novelty And Reproducibility:**

The paper is well written. However, a more clear and centralized definition of the method, its assumptions, and its limitations would be useful. The results seem to be novel, high quality, and reproducible.

**Strength And Weaknesses:**

STRENGTHS:
* The paper studies an important problem. Catastrophic forgetting is a well-known problem in the field, and the method has applications to other areas as well.
* The algorithmic ideas proposed in the work are novel, technically sound, and non-trivial. The depth of the technical results and their derivations is significant.
* The experiments are extensive and look at a wide range of interesting settings: influence functions, continual learning, etc.

WEAKNESSES:
* I think some of the tasks used some experiments can be made more realistic. The influence functions experiments look at a few toy UCI datasets. I'm not an expert in continual learning, but the main datasets seem versions of MNIST as well.
* I think the presentation can be further improved by better explaining the limitations of the method relative to existing approaches. Additionally, the authors discuss extensions to non-fully-connected architectures in the paper, but I feel like most of the details are in the appendix. A clear explanation of the class of models to which this does or does not apply would help.

**Summary Of The Paper:**

The authors propose a way to approximate the function space distance for certain broad classes of neural networks. This methods is less computationally expensive and more precise than alternatives based on the Taylor decomposition of the FSD, and are particularly suited for continual learning.

**Summary Of The Review:**

I can see this paper being a useful addition to this line of literature.

---

> ### Author Response · Authors · 2022-11-12
> **Response to Reviewer cRKz**
>
> Thank you to the reviewer for the encouragingly positive review and for highlighting the importance of the problem of catastrophic forgetting in several applications. We appreciate that our algorithm was found to be “technically sound” and evaluated with extensive experiments. Below, we address each of the concerns and incorporate suggestions to improve the presentation in the revision.
>
> >I think some of the tasks used some experiments can be made more realistic. The influence functions experiments look at a few toy UCI datasets. I'm not an expert in continual learning, but the main datasets seem versions of MNIST as well.
> - For influence function experiments, we evaluate our method on UCI tasks in Section 5.3 and also apply it to a mislabeled example identification task in a classification setting, details for which are included in Appendix F. In the latter, we demonstrate that BGLN matches the performance of the (ground-truth) PBRF objective, while avoiding the iterative procedure over the full dataset that PBRF depends on. For continual learning, we follow prior works [1, 2] and evaluate BGLN on the benchmarks therein.
>
> > I think the presentation can be further improved by better explaining the limitations of the method relative to existing approaches. A more clear and centralized definition of the method, its assumptions, and its limitations would be useful.
> - We thank the reviewer for this suggestion. We have accordingly revised Section 6 to include a discussion of the limitations of our methods. While they make a closer approximation to the true FSD in principle, they are limited by independence and probabilistic modeling assumptions. We would also like to point the reviewer to writing improvements made in Sections 3.1 and 3.2 which present our method and its assumptions more completely and cohesively.
>
> >Additionally, the authors discuss extensions to non-fully-connected architectures in the paper, but I feel like most of the details are in the appendix. A clear explanation of the class of models to which this does or does not apply would help.
> - While low-level details for convolutional architectures are deferred to Appendix C, we clearly state the applicability of our methods to any linear network with nonlinear activations at the end of Section 3.1 of the revised draft. We would also like to note that the discussion in Sections 3.1 and 3.2 is general and applies to both fully-connected and convolutional networks.
>
> We hope that these updates and clarifications fully address the reviewers’ comments and once again, appreciate the support for our paper. Please let us know if there are any points we can further clarify.
>
> [1] Pingbo Pan, Siddharth Swaroop, Alexander Immer, Runa Eschenhagen, Richard Turner, and Mohammad Emtiyaz E Khan. Continual deep learning by functional regularisation of memorable past. Advances in Neural Information Processing Systems, 33:4453–4464, 2020.
>
> [2] Tim G. J. Rudner, Freddie Bickford Smith, Qixuan Feng, Yee Whye Teh, and Yarin Gal. Continual Learning via Sequential Function-Space Variational Inference. In Proceedings of the 39th International Conference on Machine Learning, Proceedings of Machine Learning Research. PMLR, 2022.

---

### Official Review · Reviewer_qY8f · 2022-10-25

**Confidence:** 3
**Correctness:** 2
**Technical Novelty And Significance:** 2
**Empirical Novelty And Significance:** 2
**Recommendation:** 5

**Clarity, Quality, Novelty And Reproducibility:**

The paper is very well written and easy to follow. I am not convinced there is enough mathematical novelty in the solution provided, since it makes two seemingly arbitrary choices of linearisation and independent Bernoullis without any justification or analysis. The experiments appear to be adequately reproducible.

**Strength And Weaknesses:**

**Strengths:**
- The problem considered is a worthy one.
- The paper is very clearly presented, well-written, and conveys and elegant message.

**Weaknesses:**
- Section 3.1. What does linearisation (5-10) represent, and where does it come from? It is not a first-order Taylor in the inputs, since the Taylor series of the ReLU does not converge to the ReLU.
- "The upshot of our approximation is that, in the ReLU case, it only depends on the training data through the signs of all the activations (since passing layer inputs through $\phi$ and $\phi'$ both result in a multiplication by 0 or 1)". I am not understanding something here. I thought it would depend on the sign of the *preactivations* $s$. Additionally, passing through $\phi$ involves multiplication by zero or one, but passing through $\phi'$ involves taking the sign of the input.
- Beyond computational advantage, why is the step function approximated using an independent Bernoulli random variable? This changes the dynamic. Does this modification preserve the original formulation in any sense?
- Given that this paper contains no mathematical theory, I would have hoped to see more realistic datasets for baselines beyond MNIST, CIFAR, Wine, Housing, etc.

**Summary Of The Paper:**

**Post rebuttal summary**

I thank the authors for their thoughtful responses. This well-written paper tackles the important problem of FSD estimation. The methodology introduced is not accompanied by a sufficiently convincing theoretical or empirical analysis. The updated empirical analysis hints that this method might be worth further exploring more thoroughly beyond isolated examples. I encourage the authors to submit an improved version of the paper in future.

======

Summarising the properties of a dataset (perhaps with respect to a model) finds useful application in problems such as continual learning, domain adaptation, semi-supervised learning, influence-function estimation and even regularisation. In particular, it may be used to avoid catastrophic forgetting. The authors use function space distance (FSD) for this purpose, applied to ReLU neural networks. Using a certain linear approximation and stochastic Bernoulli approximation of the FSD, they derive an efficient algorithm for approximating the FSD. They find their algorithm to be competitive with SOTA on some toy examples.

**Summary Of The Review:**

While well-written, I am not convinced that this paper contains enough insight to be published.

It would be okay for the benchmark datasets to be somewhat toy *if* the mathematical formulation, motivation and analysis were complete. At the moment, I can see two unprincipled approximations: (1) The "linearisation" in input space, and (2) the replacement of a step function with an independent(!) Bernoulli random variable. Alternatively, if the benchmarks were more convincing, the mathematical analysis could be acceptably less complete.

---

> ### Author Response · Authors · 2022-11-12
> **Response to Reviewer qY8f**
>
> We thank the reviewer for their summary of our work and contributions, highlighting the problem setting as “worthy” and describing the paper as “very clearly presented, well-written, and conveys and elegant message”. We appreciate the feedback and suggestions for improvement, each of which we address below.
>
> >Section 3.1. What does linearisation (5-10) represent, and where does it come from? It is not a first-order Taylor in the inputs, since the Taylor series of the ReLU does not converge to the ReLU.
> - In Section 2, we reinterpret previous parametric approximations to the FSD as second-order Taylor approximations, or equivalently, network linearization [1]. It is due to this interpretation that we motivate and propose a more fine-grained linearization of each step of the computation in the network with respect to its inputs, to more accurately approximate the FSD without incurring significant memory costs.
>
> >"The upshot of our approximation is that, in the ReLU case, it only depends on the training data through the signs of all the activations (since passing layer inputs through ϕ and ϕ′ both result in a multiplication by 0 or 1)". I am not understanding something here. I thought it would depend on the sign of the preactivations s. Additionally, passing through ϕ involves multiplication by zero or one, but passing through ϕ′ involves taking the sign of the input.
> - Thank you to the reviewer for pointing out the inaccurate wording. We have revised this sentence in Section 3.2 to consider the signs of preactivations. We further clarify that passing layer inputs through $\phi$ and multiplying by the derivative $\phi'$, both result in a multiplication by 0 or 1 depending on the preactivation signs.
>
> >Beyond computational advantage, why is the step function approximated using an independent Bernoulli random variable? This changes the dynamic. Does this modification preserve the original formulation in any sense?
> - The use of Bernoulli random variables is motivated by the above-mentioned observation that activation computation depends on a multiplication factor of 0 or 1. When considering moments of the $a$ and $\Delta a$, this multiplication factor corresponds to the MLE of a Bernoulli random variable and hence, can be modeled as such. While this modeling could change the outputs of individual networks, we are concerned with the expected FSD, in which case it is well-motivated as above. Taking this reviewer’s comments into account, we make this motivation clearer at the start of Section 3.2. Note that the independence of these variables is indeed an assumption that our method makes for ease of computation, and we empirically test its validity by measuring the closeness of our estimate to the true FSD and evaluating it on a variety of tasks and settings in Section 5.
> - We believe that this, along with the motivation for linearized activation functions discussed above, justifies our approximations and choices well, which our further corroborated by experimental evaluation.
>
> We hope that we have satisfactorily clarified each of the reviewer’s questions and highlighted the insights in our contributions, and that they would consider an increased rating of our work based on these. Please let us know if there are any other points we can clarify.
>
>
> [1] Roger Grosse. University of Toronto CSC2541, Topics in Machine Learning: Neural Net Training Dynamics, Chapter 4: Second-Order Optimization. Lecture Notes, 2021.

---

> > ### Comment · Reviewer_qY8f · 2022-11-17
> > **Response to authors**
> >
> > Thanks for thoughtfully responding to my feedback.
> >
> > > In Section 2, we reinterpret previous parametric approximations to the FSD as second-order Taylor approximations,...
> >
> > It is not the interpretation of approximations to FSD as second-order Taylor expansions that I am concerned with. It is that if the network involved contains ReLU activations, I wonder how can the Taylor expansion be meaningful? Even an infinite order Taylor expansion will not be equivalent to the network itself, since the Taylor expansion of the ReLU does not converge.
> >
> > > While this modeling could change the outputs of individual networks, we are concerned with the expected FSD, in which case it is well-motivated as above...Note that the independence of these variables is indeed an assumption that our method makes for ease of computation, and we empirically test its validity by measuring the closeness of our estimate to the true FSD and evaluating it on a variety of tasks and settings in Section 5...
> >
> > I am not convinced through theory nor experiment that this independent Bernoulli model accurately models ReLU networks. You mention that it is well-motivated, but I cannot see any theoretical description of the error incurred by this assumption, or even a satisfying heuristic motivation. The empirical evaluation is on CIFAR10, Split MNIST, permuted MNIST, and does not give confidence that the method generalises to other situations.
> >
> > I reiterate that I find the paper very well-written, but unfortunately it lacks thorough theoretical and empirical analysis.

---

> > > ### Author Response · Authors · 2022-11-18
> > > **Further response to Reviewer qY8f**
> > >
> > > Thank you, we appreciate your response! While we acknowledge that the paper does not present a formal theoretical analysis, we believe that our empirical analyses demonstrate the effectiveness of our approach.
> > >
> > > - We would like to highlight that we evaluate our method on Split MNIST, Permuted MNIST, and Split **CIFAR100**, which are standard and comprehensive benchmarks in continual learning literature (e.g. [1, 2]). Please note that Split CIFAR100 is considered a challenging task in continual learning.
> > > - We were able to obtain competitive results competitive with the current SOTA on MNIST with **significantly** less memory requirement compared to prior works.
> > > - We have added empirical results on comparing FSD estimates to the true FSD on CIFAR100 tasks in Figure 4 of Appendix D. On this task, we show that the BGLN-S aligns with the true FSD better than previous approximations such as NTK and EWC.
> > > - Moreover, we evaluate the usefulness of our approach on influence function estimation tasks and demonstrate that our approach is not limited to the continual learning set-up.
> > >
> > > [1] Pingbo Pan, Siddharth Swaroop, Alexander Immer, Runa Eschenhagen, Richard Turner, and Mohammad Emtiyaz E Khan. Continual deep learning by functional regularisation of memorable past. Advances in Neural Information Processing Systems, 33:4453–4464, 2020.
> > >
> > > [2] Tim G. J. Rudner, Freddie Bickford Smith, Qixuan Feng, Yee Whye Teh, and Yarin Gal. Continual Learning via Sequential Function-Space Variational Inference. In Proceedings of the 39th International Conference on Machine Learning, Proceedings of Machine Learning Research. PMLR, 2022.

---

### Official Review · Reviewer_i4TZ · 2022-10-25

**Confidence:** 4
**Correctness:** 3
**Technical Novelty And Significance:** 2
**Empirical Novelty And Significance:** 2
**Recommendation:** 5

**Clarity, Quality, Novelty And Reproducibility:**

Overall, the paper is clear. I would suggest that the authors improve upon the following two points:

- In Figure 3a, a different shape should be used for each method, as otherwise it is very difficult to read.

- I would be grateful if more details could be given in Table 1 about each of the proposed methods, and especially about the class-conditioned (CW) variants.

Regarding the novelty, I think that the idea behind the work is interesting, but the work itself does not make a convincing enough argument about its usefulness.

Regarding reproducibility, the authors plan to release their code, and provide further training details in the Appendix.

**Strength And Weaknesses:**

Strengths:

- The method proposed is very easy to understand and can be easily implemented. The authors measure function distance by essentially substituting the ReLU functions of the network with stochastic gating variables. Implicitly, this process samples from the various linear regions of the network, making the calculation of distance between them straightforward.

- The authors examine an interesting setting for the application of their function space distance approximation in continual learning. In this setting, the authors use their FSD approximation to train models that sequentially learn new tasks, while aiming to retain accuracy in the previous ones.

Weaknesses:

- One major weakness of the proposed FSD approximation technique is that there is not sufficient evidence that it is a good approximation of the distance between two networks. In order to prove this theoretically, there would need to be proofs of bounds for the FSD estimated via the proposed method compared to the true FSD, or at the very least a simple toy example demonstrating the FSD estimated on a given network, on a toy dataset. To prove this experimentally, the authors would need to include further comparisons like those included in Figure 3a. Currently, this comparison between estimated and true FSD is only done on an MNIST variant, so doing so over more complicated datasets would be helpful to properly evaluate how well the proposed method performs. Including either of the two would greatly improve the paper.

- In a similar vein, the experimental results on continual learning are not convincing enough for the capabilities of the proposed method. More specifically, the proposed methods only show performance benefits in the case of MNIST variants, and not on the CIFAR100 continual learning benchmark. This, combined with the previous result, makes me wonder whether the approximation derived from the proposed method is useful for the desired continual learning task. I believe the authors would need to further analyze why their method underperforms in the CIFAR100 variant.

- Finally, the authors state that their method can approximate the FSD between two networks without expensive storage requirements, compared to other nonparametric methods. I believe that these storage comparisons should be included (for example, in Table 1), in order to fully understand the benefits of the proposed method, from a memory requirement perspective.

Update post rebuttal: The authors have included extra experiments that somewhat alleviate my concerns regarding the quality of the FSD approximation. The proposed method seems to outperform NTK in Split-CIFAR100, as far as FSD approximation is concerned. Moreover, the authors have included explicit memory requirements for their proposed methods.

However, the experimental part is still inconclusive, in the sense that while there is an experimental indication that the proposed method performs well with respect to estimating the FSD, it lags behind NTK on the same problem. Furthermore, while there is theoretical motivation with respect to the gating units used, it hinges on these being independent (as pointed out by another reviewer), which seems to be a very limiting assumption. Including a simple toy example where this choice of gating is the best one would be a good addition to the paper.


**Summary Of The Paper:**

This paper proposes a method to estimate the function space distance (FSD) of two networks. This method relies on transforming the ReLU layers of the network into Bernoulli stochastic gates, and measuring the distance between the networks while sampling from these gating variables. Using this method, the authors are able to estimate the function distance between two networks, and can use this technique for tasks where this function distance is required, such as in continual learning.

**Summary Of The Review:**

I think this work presents an interesting idea, but given that there is not sufficient evidence that the FSD approximation proposed performs well in general, and the main results do not make a convincing argument for the usefulness of the method (other than memory requirements, which are not fully elaborated on), I believe that it needs to be improved.

Update post-rebuttal: I have raised my score following the authors' responses, as some of my concerns have been alleviated. I believe that the paper can still be improved, by further examining why the proposed method lags behind NTK in performance (despite experimentally seeming to better approximate the FSD) and including further theoretical justification on why the gating units chosen are a good choice.

---

> ### Author Response · Authors · 2022-11-12
> **Response to Reviewer i4TZ**
>
> We thank the reviewer for their summary of our paper and highlighting its simplicity and application to a variety of interesting settings. We appreciate the recommendations for improvement and address each of the concerns brought up below.
>
> >One major weakness of the proposed FSD approximation technique is that there is not sufficient evidence that it is a good approximation of the distance between two networks. In order to prove this theoretically, there would need to be proofs of bounds for the FSD estimated via the proposed method compared to the true FSD, or at the very least a simple toy example demonstrating the FSD estimated on a given network, on a toy dataset. To prove this experimentally, the authors would need to include further comparisons like those included in Figure 3a. Currently, this comparison between estimated and true FSD is only done on an MNIST variant, so doing so over more complicated datasets would be helpful to properly evaluate how well the proposed method performs. Including either of the two would greatly improve the paper.
> - We agree with the reviewer’s suggestion to corroborate our intuition for more complicated datasets, and include a comparison of FSD estimates for networks trained on Split CIFAR100. To directly compare the NTK approximation with the linearized activation function trick, we compute the estimated FSD between pairs of networks using these two kinds of linearization, when provided with the same information, and plot them against the true empirical FSD. Figure 4 added in Appendix D shows that BGLN-S estimates correlate better with the true FSD than NTK, hence corroborating our intuition about linearizing activation functions. To quantify this difference, we also measure the Spearman rank-order and Kendall's Tau correlation coefficients for each estimator with the true FSD. BGLN-S obtains values 96.36 and 79.19, respectively, outperforming NTK, which obtains 86.42 and 71.71, respectively.
> - Further, we would like to reemphasize our experimental results on influence function estimation in Section 5.3, which was included with the purpose of evaluating our method across a breadth of tasks and settings. In the case of influence functions, the true FSD is equivalent to the PBRF objective, as derived in [1]. Table 4 shows a very high correlation between influence estimates given using BGLN and the true FSD as measured by Pearson and Spearman rank-order correlation, across datasets, outperforming comparable approaches.
>
> >I believe the authors would need to further analyze why their method underperforms in the CIFAR100 variant.
> - We note that while our methods make a closer approximation to the true FSD in principle, they are limited by independence and probabilistic modeling assumptions. Empirically, these may explain its lower but competitive performance on Split CIFAR100. Interestingly, the direct NTK approximation outperforms all previous methods, which makes optimizing its memory-efficiency an attractive direction for future work. We include these limitations of our work in Section 6 of the revision.
>
> >I believe that these storage comparisons should be included (for example, in Table 1), in order to fully understand the benefits of the proposed method, from a memory requirement perspective. I would be grateful if more details could be given in Table 1 about each of the proposed methods, and especially about the class-conditioned (CW) variants.
> - In order to make the advantages of our method more explicit, we follow this suggestion and include details about memory cost analysis and comparison in Table 1 and Appendix E.1 of the revised draft. The equations therein clearly establish that our method and its variants incur only constant memory cost in terms of data to be stored. This can be contrasted with other high-performing approaches, for both continual learning and influence function estimation. In the former, nonparametric methods have memory costs that scale (at least) linearly depending on the coreset size, which can be in the hundreds. In the latter, other methods require iterating over the entire training dataset, whose size is generally at least in the thousands.
>
> >In Figure 3a, a different shape should be used for each method, as otherwise it is very difficult to read.
> - We appreciate this suggestion to improve the readability and presentation of our results. Figure 3a in the revised draft is accordingly updated.
>
> We hope that the reviewer finds our clarifications and updates to adequately address their questions and considers a higher recommendation of our work. Please let us know if there are any other points we can clarify.
>
> [1] Juhan Bae, Nathan Ng, Alston Lo, Marzyeh Ghassemi, and Roger Grosse. If influence functions are the answer, then what is the question? arXiv preprint arXiv:2209.05364, 2022a

---

> > ### Comment · Reviewer_i4TZ · 2022-11-17
> > **Thank you for your responses.**
> >
> > Thank you very much for the responses to my comments.
> >
> > I read the updated draft, and I can see that additional comparisons have been included regarding the FSD on CIFAR100, as well as the memory costs comparison in Table 1. These are both improvements upon the previous version of the paper, and as such I have raised my score.
> >
> > Regarding the theory, I understand the issues arising from independence which are mentioned as a possible reason for the lower results on Split CIFAR100 (and which are also raised by Reviewer qY8f). I appreciate the extra elements added regarding the motivation of using Bernoulli random variables as MLE estimators for the activations, however some simple theoretical analysis on how well this estimate performs would be useful for this paper, in order to make a more convincing argument.

---

### Official Review · Reviewer_QzNA · 2022-10-26

**Confidence:** 3
**Correctness:** 3
**Technical Novelty And Significance:** 2
**Empirical Novelty And Significance:** 2
**Recommendation:** 6

**Clarity, Quality, Novelty And Reproducibility:**

Clarity: the clarity needs to be improved, as discussed in weaknesses.

Quality: the quality of the BGLN algorithm is fine. However, the advantage of competitive approaches should be make more clearly.

Novelty: as far as I know, the idea of approximating FSD with moments is novel.

Reproducibility: code is not submitted. There is a pseudocode in the supplementary material. I believe the reproducibility is generally ok.

**Strength And Weaknesses:**

Strength:
- The proposed approach is quite simple and accurate.

Weaknesses:
- The presentation needs to be involved. The current paper has some logical discontinuities. For example, at the end of Sec. 3.1, there misses a part on how to compute the FSD with the activation moments. There should be some derivations and pseudocode for that. Otherwise, the discussions in Sec. 3.2 and Sec. 3.3 seem somewhat random.

Sec. 2 might also be somewhat disruptive for the readers might not understand what is parametric and nonparametric approaches here. It might be better to put the concrete formulation of continual learning in Sec. 5.1 forward to Sec. 2 to act as a motivating example.

There is some typos: for example, in Eq. (3), some (l) should be in fact (l-1).

- The advantage of linearizing the activation over competitive approaches should be discussed in more detail. Currently, there is only empirical evidences of the accuracy of BGLN, which is not strong enough. The paper would be more solid if there is some error / bias analysis and comparison.

**Summary Of The Paper:**

Post rebuttal
====

Thank the authors for the responses. Since the responses addressed my original concerns, I raised my score. However, I agree with reviewer i4TZ and qY8f that the theoretical and empirical studies are still somewhat simple, and could still be improved.



This paper considers how to obtain a parametric approximation to the empirical function space distance (FSD) of two networks, i.e., $\mathbb E_{p_{emp}}[\rho(f(x, \theta_0), f(x, \theta_1))]$. The key assumption here is we cannot store the training data $p_{emp}$, as it is a  infinite stream. On the other hand, the overhead of storing the model is acceptable.

The paper takes a linearization approach, BGLN, which approximates the activation of the second network $\theta_1$ with a first-order Taylor's expansion at the activation of the first network $\theta_0$. By the linearity of expectation, the FSD can be written with the activation gradient's moment: $\mathbb E_{p_{emp}}[\phi^\prime(a_0)]$. By computing and storing such moments (i.e., sufficient statistics) in advance, the FSD can be computed without accessing the data distribution $p_{emp}$.

The proposed BGLN is empirically shown to be more accurate than competitive approaches, such as second-order approximation to the FSD on the parameter space or NTK-based approximation, which again linearizes on the parameter space. The effectiveness of the proposed approach is shown on continual learning and influence function estimation tasks.

**Summary Of The Review:**

The paper proposes a reasonable solution to a practical problem of function space distance estimation. However, presentation and relationships with existing works should be improved. The significance of the paper is not clear in its current form.

---

> ### Author Response · Authors · 2022-11-12
> **Response to Reviewer QzNA (1/2)**
>
> Thank you to the reviewer for their succinct summary of our work and for describing it as “simple”, “accurate” and “novel”. Below, we address each of their concerns and take into account their suggestions for a clearer and better structured revision.
>
> >The key assumption here is we cannot store the training data pemp, as it is a infinite stream. On the other hand, the overhead of storing the model is acceptable.
> - We would like to clarify that one copy of the model parameters contributes to a storage cost that is common across all approaches to the applications we consider. The primary memory gains that our method enjoys are due to reducing additional parameter information and eliminating coresets of data samples to be stored.
>
> >The presentation needs to be involved. The current paper has some logical discontinuities. For example, at the end of Sec. 3.1, there misses a part on how to compute the FSD with the activation moments. There should be some derivations and pseudocode for that. Otherwise, the discussions in Sec. 3.2 and Sec. 3.3 seem somewhat random.
> - We have revised the structure of Sections 3.1 and 3.2 to give a more complete and logical presentation of our methods. In Section 3.1, we discuss the general linearized activation function trick and motivate its advantages over the common parameter space linearization approaches. Section 3.2 describes the stochastic gating strategy for ReLU networks, the information we need to store, and the propagation of the data moments through the network to compute FSD. We refer to the pseudocode in Algorithm 1, which makes these steps explicit. Having discussed these strategies, Section 3.3 puts these parts together to describe the final algorithm, with exact equations for our deterministic variant deferred to Appendix B.
>
> > Sec. 2 might also be somewhat disruptive for the readers might not understand what is parametric and nonparametric approaches here. It might be better to put the concrete formulation of continual learning in Sec. 5.1 forward to Sec. 2 to act as a motivating example.
> - We appreciate this suggestion to better motivate the relevance of FSD approximation via the example of continual learning. The revised paper now contains this motivating example in Section 2. We also make the notation and definition of FSD in Equations 1 and 2 clearer by making its dependence on the data distribution explicit. We would also like to highlight that paragraph 2 of the introduction in Section 1 explains the difference between parametric and nonparametric approaches and discusses how their performance compares.
>
> >There is some typos: for example, in Eq. (3), some (l) should be in fact (l-1).
> - Thank you for pointing out this typo! Equation 3 has been accordingly fixed in the revision.
>
> > Quality: the quality of the BGLN algorithm is fine. However, the advantage of competitive approaches should be make more clearly.
> - Since our main contribution is a memory-efficient parametric method that is competitive in performance with more memory-intensive nonparametric counterparts, we include details of memory cost analysis in Table 1 and Appendix E.1. This clearly demonstrates the memory advantage our methods enjoy.
> - We also include the limitations of our method, as compared to approaches that assume access to more information, in Section 6. We note that while our methods make a closer approximation to the true FSD in principle, they are limited by independence and probabilistic modeling assumptions.

---

> ### Author Response · Authors · 2022-11-12
> **Response to Reviewer QzNA (2/2)**
>
> >The advantage of linearizing the activation over competitive approaches should be discussed in more detail. Currently, there is only empirical evidences of the accuracy of BGLN, which is not strong enough. The paper would be more solid if there is some error / bias analysis and comparison.
> - We highlight two advantages BGLN has over methods based on parameter space linearization. First, our linearization is with respect to inputs instead of parameters, hence capturing nonlinear interactions between the parameters in different layers. Second, the only computations that introduce linearization errors into our approximation are those involving activation functions, in contrast with other methods which suffer linearization errors for each layer, including linear layers, where nonlinear parameter dependencies exist. This is now more clearly stated in Section 3.1 in the revision.
> - More formally, we can analyze the error of BGLN-S as compared to the true FSD, propagated through each step of the network’s computation. In the case of linear (fully-connected or convolutional) networks with continuously differentiable activation functions, the error introduced due to activations is the second-order Taylor error, $err(a^{(l)}) = \mathcal{O}(||s_1^{(l)} - s_0^{(l)}||^2)$. Any subsequent computation then has error only due to $err(a^{(l)})$, with no further errors introduced due to linear layers. For instance, for fully-connected layers of the two networks parametrized using $W_0^{(l)}$ and $W_1^{(l)}$, which have as inputs $a_0^{(l-1)}$ and $a_1^{(l-1)}$, the FSD estimation error is equal to $W_1^{(l)} err(a^{(l-1)})$, which only depends on activation error in the previous step and introduces no new errors. Full definitions and derivations for these equations are included in Appendix G.
> - We have strengthened our empirical verification of this benefit by comparing the closeness of FSD estimates given by BGLN-S and the NTK approximation to the true empirical FSD on the more complicated Split CIFAR100 dataset. Figure 4 added in Appendix D shows that BGLN-S estimates correlate better with the true FSD than NTK, hence corroborating our intuition about linearizing activation functions. To quantify this difference, we also measure the Spearman rank-order and Kendall's Tau correlation coefficients for each estimator with the true FSD. BGLN-S obtains values 96.36 and 79.19, respectively, outperforming NTK, which obtains 86.42 and 71.71, respectively.
>
> We hope that the above satisfactorily addresses each of the reviewer’s concerns and warrants an increased rating. Please let us know if there are any other points we can clarify.

---

### Author Response · Authors · 2022-11-12
**General summary of clarifications and revisions**

We are grateful the reviewers found our work simple to understand and implement (reviewers QzNA, i4TZ), well-written (reviewers qY8f, cRKz), studying an important problem (reviewers qY8f, cRKz), applied to interesting settings (reviewers i4TZ, cRKz) and novel (reviewers QzNA, cRKz). We would like to thank all the reviewers for their comments and constructive feedback which have improved our paper! We have addressed each concern of each reviewer separately in individual responses to the reviews. Corresponding updates to the paper can be found highlighted in ${\color{blue} \text{blue}}$ in the revision draft. Here, we summarize the main clarifications and revisions.

1. Method Section Restructuring
- We have revised the structure of Sections 3.1 and 3.2 to give a more complete and coherent presentation of our methods.
- In Section 3.1, we discuss the general linearized activation function trick and motivate its advantages over the common parameter space linearization approaches.
- Section 3.2 describes the stochastic gating strategy for ReLU networks, the information we need to store, and the propagation of the data moments through the network to compute FSD.

2. Memory Cost Analysis
- In order to make the advantages of our method more explicit, we include details about memory cost analysis and comparison in Table 1 and Appendix E.1 of the revised draft. The equations therein clearly establish that our method and its variants incur only constant memory cost in terms of data to be stored.
- This can be contrasted with other high-performing approaches, for both continual learning and influence function estimation. In the former, nonparametric methods have memory costs that scale (at least) linearly with the coreset size, which can be in the hundreds. In the latter, other methods require iterating over the entire training dataset, whose size is generally at least in the thousands.

3. Comparing FSD Estimators for Split CIFAR100
- We include a comparison of FSD estimates for networks trained on Split CIFAR100, with the purpose of demonstrating the closeness of BGLN to the true FSD on a more complicated dataset. To directly compare the NTK approximation with the linearized activation function trick, we compute the estimated FSD between pairs of networks using these two kinds of linearization, when provided with the same information, and plot them against the true empirical FSD.
- Figure 4 added in Appendix D shows that BGLN-S estimates correlate better with the true FSD than NTK, hence corroborating our intuition about the benefits of linearizing activation functions.
- To quantify this difference, we also measure the Spearman rank-order and Kendall's Tau correlation coefficients for each estimator with the true FSD. BGLN-S obtains values 96.36 and 79.19, respectively, outperforming NTK, which obtains 86.42 and 71.71, respectively.

4. Limitations
- We include the limitations of our method, as compared to approaches that assume access to more information, in Section 6.
- We note that while our methods make a closer approximation to the true FSD in principle, they are limited by independence and probabilistic modeling assumptions.

5. Applicability
- We would like to stress that our method is applicable to any linear (fully-connected or convolutional) architecture with nonlinear activations.
- Further, it can be applied successfully to a variety of problem settings that rely on FSD computation. We demonstrate this application in the two settings of continual learning and influence function estimation, where our method has lower computation and memory requirements.

---

### Author Response · Authors · 2022-11-17
**Following up on review responses**

Dear reviewers,

Thank you again for your time and effort in reviewing our paper. We greatly appreciate your comments and suggestions.

We would like to kindly remind you that we have under two days left until this discussion period ends. We believe that we successfully clarified and addressed your concerns, questions and suggestions and have incorporated them in the updated manuscript.

In case you have any further concerns or questions, please do not hesitate to let us know.

Authors

---

### Decision · Program_Chairs · 2023-01-20

**Decision:**

Reject

**Justification For Why Not Higher Score:**

There are unjustified approximations and the empirical performance improvement is not clear compared to existing methods.

**Justification For Why Not Lower Score:**

This is a new approach for an important problem.

**Metareview: Summary, Strengths And Weaknesses:**

The paper proposes a new method to estimate the functional distance between two networks, a problem of great interest in the transfer / few-shot learning literature.

All reviewers appreciated the idea and its computational qualities. They had some doubts, however, on the empirical performance and the approximations made. We had further interactions with the reviewers as the consensus was that, even though the paper shows some promise, it could still be improved.

Given the total volume of papers published, I am favoring works that address these questions completely, given that the partial work can be advertised anyway and that a complete analysis will make it easier for researchers to fully understand the landscape.

As such, I vote for rejection.

**Summary Of Ac-Reviewer Meeting:**

We did not have a meeting since the two reviewers who engaged the most replied in writing with their points.